# Systematic benchmarking of high-throughput subcellular spatial transcriptomics platforms across human tumors

Pengfei Ren[1,2,10], Rui Zhang[1,10], Yunfeng Wang[1,2], Peng Zhang[2], Ce Luo[2], Suyan Wang[3], Xiaohong Li[2], Zongxu Zhang[2], Yanping Zhao[4], Yufeng He[1], Haorui Zhang[1], Yufeng Li[5], Zhidong Gao[6], Xiuping Zhang[7], Yahui Zhao[8], Zhihua Liu[8], Yuanguang Meng[9] ✉, Zhe Zhang[9] ✉ & Zexian Zeng[1,2,3] ✉

Recent advancements in spatial transcriptomics technologies have significantly enhanced resolution and throughput, underscoring an urgent need for systematic benchmarking. Here, we generate serial tissue sections from colon adenocarcinoma, hepatocellular carcinoma, and ovarian cancer samples for systematic evaluation. Using these uniformly processed samples, we generate spatial transcriptomics data across four high-throughput platforms with subcellular resolution: Stereo-seq v1.3, Visium HD FFPE, CosMx 6K, and Xenium 5K. To establish ground truth datasets, we profile proteins on tissue sections adjacent to all platforms using CODEX and perform single-cell RNA sequencing on the same samples. Leveraging manual nuclear segmentation and detailed annotations, we systematically assess each platform's performance across capture sensitivity, specificity, diffusion control, cell segmentation, cell annotation, spatial clustering, and concordance with adjacent CODEX. The uniformly generated and processed multi-omics dataset could advance computational method development and biological discoveries. The dataset is accessible via SPATCH, a user-friendly web server for visualization and download.

Spatially resolved transcriptomics integrates high-throughput transcriptomic profiling with spatially contextualized tissue architecture, bridging the gap in single-cell RNA sequencing (scRNA-seq) by linking molecular profiles to their spatial context. With preserved spatial information, this technology offers unprecedented insights into cellular states, intercellular interactions, and tissue organization[1–6]. Its applications span multiple biological disciplines: in neuroscience, it enables high-resolution mapping of neural circuits and molecular

[1]Peking-Tsinghua Center for Life Sciences, Academy for Advanced Interdisciplinary Studies, Peking University, Beijing, China. [2]Center for Quantitative Biology, Academy for Advanced Interdisciplinary Studies, Peking University, Beijing, China. [3]Peking University Chengdu Academy for Advanced Interdisciplinary Biotechnologies, Chengdu, Sichuan, China. [4]Tsinghua-Peking Center for Life Sciences, School of Life Sciences, Tsinghua University, Beijing, China. [5]Department of Obstetrics and Gynecology, Chinese PLA Medical School, Beijing, China. [6]Department of Gastroenterological Surgery, Peking University People's Hospital, Beijing, China. [7]Faculty of Hepato-Biliary-Pancreatic Surgery, the First Medical Center of Chinese People's Liberation Army (PLA) General Hospital, Chinese PLA Medical School, Beijing, China. [8]State Key Laboratory of Molecular Oncology, National Cancer Center, National Clinical Research Center for Cancer, Cancer Hospital, Chinese Academy of Medical Sciences and Peking Union Medical College, Beijing, China. [9]Department of Obstetrics and Gynecology, Seventh Medical Center of Chinese PLA General Hospital, Beijing, China. [10]These authors contributed equally: Pengfei Ren, Rui Zhang. ✉e-mail: meng6512@sina.vip.com; tj.zhe.zhang@gmail.com; zexianzeng@pku.edu.cn

connectivity; in developmental biology, it illuminates the molecular mechanisms underlying tissue morphogenesis; and in cancer biology, it provides detailed characterization of tumor microenvironments and immune landscapes[7–10]. Driven by its transformative potential, spatial transcriptomics (ST) technology has undergone rapid development and innovation.

Spatial transcriptomics technologies can be broadly categorized into sequencing-based (sST) and imaging-based (iST) platforms, each offering distinct methodologies and advantages. sST platforms enable unbiased whole-transcriptome analysis by capturing poly(A)-tailed transcripts with poly(dT) oligos on spatially barcoded arrays. These platforms vary in capture efficiency, transcript diffusion control, and spatial resolution, ranging from microscale to nanoscale. Notable platforms include Visium[11], DBiT-seq[12], Patho-DBiT[13], Stereo-seq[14], Slide-seq[15], Slide-seqV2[16], HDST[17], sciSpace[18], PIXEL-seq[19], Seq-Scope[20], MAGIC-seq[21], and Open-ST[22]. Conversely, iST platforms utilize iterative hybridization of fluorescently labeled probes followed by sequential imaging to profile gene expression in situ at single-molecule resolution. iST technologies differ in probe design, signal amplification strategies, imaging modalities, and target genes. Notable platforms include ISS[23], CosMx[24], Xenium[25], MERFISH[26], seqFISH[27], osmFISH[28], and STARmap[29]. These complementary approaches underscore the strengths of ST, with sST providing unbiased transcriptome-wide coverage and iST offering high resolution and sensitivity for the detection of target genes.

Several efforts have been made to benchmark both sST and iST technologies. A comprehensive evaluation by Yue et al.[30] compared different sST platforms, including Stereo-seq[14] with 0.5 μm sequencing spots and Visium[11] with its 55 μm resolution. For iST platforms, Xenium, MERSCOPE, and CosMx were compared using gene panels ranging from 200 to 1000 genes[31–33]. A comparative study involving four iST platforms was conducted using in-house and public data, with gene panels reaching 345 genes[34]. Additionally, Austin et al.[35] evaluated six iST platforms using public datasets, with panel sizes ranging from 99 to 1147 genes. While these studies provide valuable insights, they primarily focus on ST technologies with lower spatial resolution or limited gene panel sizes. Furthermore, many benchmarking studies rely on public datasets generated under varying experimental conditions or with varying tissue types, which often lack consistent ground truth data for robust evaluation. As a result, existing efforts offer only a partial understanding of the latest advancements in ST technologies. This underscores the urgent need for a systematic benchmarking study conducted under unified experimental conditions to comprehensively evaluate the performance and comparative strengths of current high-resolution, high-throughput ST platforms.

Spatial transcriptomics has undergone remarkable advancements, with commercial platforms now achieving subcellular resolution and high-throughput gene detection. Among sST platforms, Stereo-seq v1.3[14] by BGI employs poly(dT) oligos to capture poly(A)-tailed RNA at a resolution of 0.5 μm. Visium HD[36] by 10x Genomics utilizes poly(dT) oligos to capture poly(A)-tailed probes targeting 18,085 genes at a resolution of 2 μm. Compared to earlier technologies, these higher resolutions facilitate more accurate profiling of individual cell transcriptional states. In parallel, iST platforms, such as CosMx 6K[24] by NanoString and Xenium 5K[25] by 10x Genomics, rely on fluorescently labeled probes and sequential imaging to profile 6175 and 5001 genes, respectively, offering single-molecule precision. Compared to earlier iST platforms such as ISS (39 genes), MERFISH (1000 genes), and STARmap (1020 genes), the substantially expanded gene panels of CosMx 6K and Xenium 5K offer enhanced resolution of cellular states, enable more comprehensive inference of intercellular communication networks, and allow for broader coverage of signaling pathway activities. Notably, the increased transcriptomic coverage also supports cross-disciplinary investigations, facilitating integrative analyses across domains such as immunology, oncology, and

neuroscience. These advancements underscore the pressing need for a systematic benchmark to enable more informed applications and continued innovation in this rapidly evolving field.

In this study, we collected clinical samples from three cancer types and generated serial tissue sections to systematically evaluate four commercially available high-throughput ST platforms with sub-cellular resolution. To establish ground truth datasets, we used CODEX to profile proteins in tissue sections adjacent to those used for each ST platform. In parallel, scRNA-seq was performed on the same samples to provide a comparative reference. We manually annotated cell types for both the scRNA-seq and CODEX data, along with nuclear boundaries in hematoxylin and eosin (H&E) and DAPI-stained images. Leveraging these comprehensive annotations, we systematically evaluated each platform's performance across critical metrics, including sensitivity, specificity, diffusion control, cell segmentation, cell annotation, spatial clustering, and transcript-protein alignment. The resulting uniformly generated, processed, and annotated multi-omics dataset, comprising 8.13 million cells, serves as a valuable resource for advancing computational method development and enabling biological discoveries. To ensure broad accessibility, we developed a user-friendly web server (SPATCH: http://spatch.pku-genomics.org/) for data visualization, exploration, and download.

## Results
### Sample preparation and multi-omics profiling
To enable a comprehensive and systematic benchmarking of ST platforms, we collected treatment-naïve tumor samples from three patients diagnosed with colon adenocarcinoma (COAD), hepatocellular carcinoma (HCC), and ovarian cancer (OV) (Supplementary Data 1). To accommodate the sample preparation requirements of each platform, we divided the tumor samples into multiple portions and processed them into formalin-fixed paraffin-embedded (FFPE) blocks, fresh-frozen (FF) blocks embedded in optimal cutting temperature (OCT) compound, or dissociated into single-cell suspensions (Fig. 1a). Serial tissue sections were uniformly generated for parallel profiling across multiple omics platforms. Detailed timelines for sample collection, fixation, embedding, sectioning, and transcriptomic profiling were documented (Supplementary Data 2).

We benchmarked four advanced ST platforms–Stereo-seq v1.3, Visium HD FFPE, CosMx 6K, and Xenium 5K–selected for their high-throughput gene capture capacity (>5000 genes), subcellular resolution (≤ 2 μm), and widespread commercial adoption (Fig. 1a). These platforms represent diverse technological strategies (Supplementary Data 3) and utilize overlapping yet distinct gene panels to capture key biological pathways (Supplementary Fig. 1a, b and Supplementary Data 4, 5). To establish comprehensive ground truth datasets for robust evaluation, we profiled proteins using CODEX on tissue sections adjacent to those used for each ST platform. In parallel, we performed scRNA-seq on matched tumor samples (Fig. 1a). The uniformly generated reference datasets enabled integrative and cross-modal comparisons across diverse platforms.

### Evaluation of molecular capture efficiency for marker genes
We first assessed the detection sensitivity of diverse cell marker genes across different ST platforms. To ensure consistent resolution across platforms and balance spatial specificity with transcript detection sensitivity, all subsequent bin-level analyses were conducted at 8 μm resolution–a biologically meaningful unit approximating the typical diameter of small immune cells. The epithelial cell marker *EPCAM* showed well-defined spatial patterns across all platforms, consistent with H&E staining and supported by Pan-Cytokeratin (PanCK) immunostaining on adjacent sections (Fig. 1b). Xenium 5K demonstrated superior sensitivity for multiple marker genes (Fig. 1c). To reduce potential biases from scanning area and tissue morphology, we restricted our analysis to regions shared across FFPE serial sections

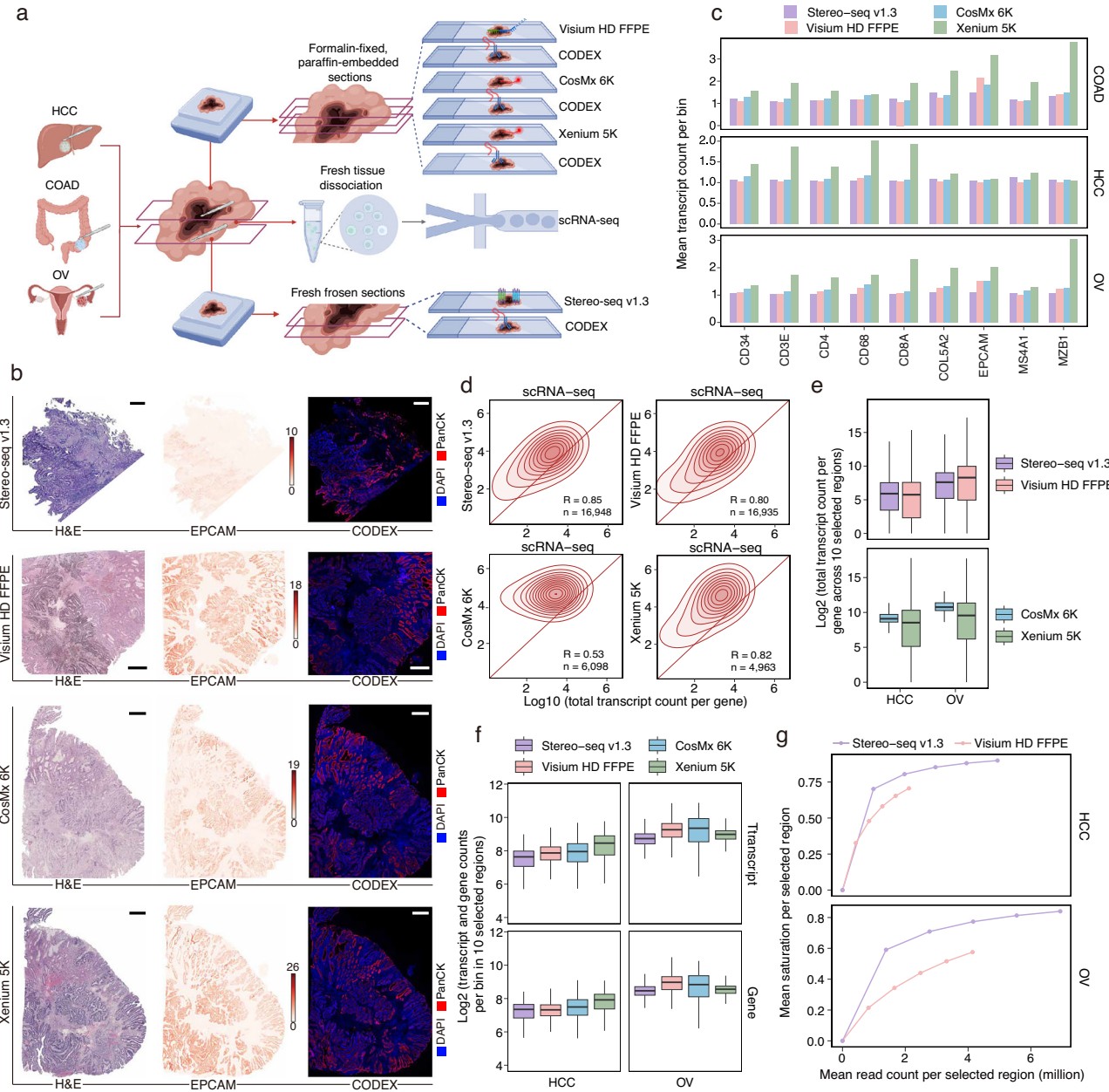

**Fig. 1 | Evaluation of gene detection sensitivity across ST platforms.**
**a** Experimental workflow. For each tumor type (COAD, HCC, OV), samples were divided into three parts: (1) FFPE blocks were used for Visium HD FFPE, CosMx 6K, and Xenium 5K; (2) fresh-frozen OCT-embedded tissue was used for Stereo-seq v1.3; (3) dissociated tissue was subjected to scRNA-seq. Sections adjacent to each ST slide were profiled by 16-plex CODEX for spatial proteomics. **b** H&E staining and EPCAM expression from ST data, along with PanCK staining from adjacent CODEX sections of COAD samples across the four ST platforms. Color intensity reflects the transcript count per 8 μm bin. Scale bars, 1 mm. **c** Mean transcript count per 8 μm bin for selected marker genes, computed across all bins with non-zero expression values over the entire tissue sections. **d** Pearson correlation of gene expression levels between ST data and scRNA-seq data. For each gene, the total transcript counts across three cancer types were averaged and $\log_{10}$ transformed. Each data point represents one gene. The diagonal red line indicates a slope of 1, and color intensity corresponds to relative gene counts. R denotes the correlation coefficient,

and n indicates the number of genes included in the analysis. **e** $\log_2$-transformed total transcript count per gene across the ten selected regions (400 × 400 μm each) in HCC and OV. Each data point represents one gene ($n = 17{,}134$ for Stereo-seq v1.3 and Visium HD FFPE, $n = 6175$ for CosMx 6K, $n = 5001$ for Xenium 5K). Center lines indicate the median value, and lower and upper hinges represent the 25th and 75th percentiles, respectively. The whiskers denote 1.5× the interquartile range. **f** $\log_2$-transformed gene and transcript counts per 8 μm bin within the ten selected regions in HCC and OV. Each data point represents one bin. Center lines indicate the median value, and lower and upper hinges represent the 25th and 75th percentiles, respectively. The whiskers denote 1.5× the interquartile range. **g** Mean sequencing saturation across the 10 selected regions for human transcripts detected by Stereo-seq v1.3 and Visium HD FFPE, calculated at stepwise increasing sequencing depths. Panel **a** created with BioRender.com. Source data are provided as a Source Data file.

(Visium HD FFPE, CosMx 6K, and Xenium 5K, Supplementary Fig. 2a). Within these shared regions, Xenium 5K consistently outperformed the other platforms (Supplementary Fig. 2b). To further reduce variability in the scanning areas, we selected ten regions of interest (ROIs, 400 × 400 µm each), primarily composed of cancer cells with similar morphology and cell density from each dataset (Supplementary Fig. 2c). Within these ROIs, we evaluated the sensitivity of cancer cell marker genes and found that Visium HD FFPE outperformed Stereo-seq v1.3, while Xenium 5K showed higher sensitivity than CosMx 6K (Supplementary Fig. 2d).

### Evaluation of molecular capture efficiency across entire gene panels

We calculated total transcript count per gene for each ST dataset and assessed their gene-wise correlation with matched scRNA-seq profiles. Stereo-seq v1.3, Visium HD FFPE, and Xenium 5K showed high correlations with scRNA-seq (Fig. 1d and Supplementary Fig. 3a). Although CosMx 6K detected a higher total number of transcripts than Xenium 5K (Supplementary Fig. 3b), its gene-wise transcript counts showed substantial deviation from matched scRNA-seq reference (Fig. 1d). This discrepancy persisted when the analysis was restricted to the 2522 genes shared between CosMx 6K and Xenium 5K (Supplementary Fig. 3c). Increasing quality control thresholds for CosMx 6K transcript calls did not significantly improve the correlation with scRNA-seq, indicating that the discrepancy is unlikely due to low-quality detections (Supplementary Fig. 3d). Cross-platform comparisons further revealed strong concordance among Stereo-seq v1.3, Visium HD FFPE, and Xenium 5K, highlighting their consistent ability to capture gene expression variation (Supplementary Fig. 3e).

To assess transcript capture across gene panels, we quantified total transcript count per gene within the ten selected ROIs from HCC and OV samples. Stereo-seq v1.3, Visium HD FFPE, and Xenium 5K exhibited comparable distributions characterized by substantial intergene variability, reflecting effective detection across a wide range of gene expression. While CosMx 6K reported higher overall transcript counts than Xenium 5K, its reduced gene-to-gene variation suggests a more limited ability to resolve differential expression (Fig. 1e). This pattern persisted when the analysis was restricted to the 2522 genes shared between CosMx 6K and Xenium 5K (Supplementary Fig. 4a). Additionally, similar patterns were observed in analyses limited to shared regions in serial FFPE sections (Supplementary Fig. 4b).

In addition to evaluating total transcript counts for individual genes, we analyzed the total numbers of transcripts and genes detected per 8 µm bin across the ten ROIs in HCC and OV samples. The COAD samples were excluded from this analysis due to their inconsistent cell density across regions. Visium HD FFPE exhibited enhanced detection capacity compared to Stereo-seq v1.3 (Fig. 1f). Xenium 5K demonstrated higher sensitivity than CosMx 6K in HCC but showed lower sensitivity in OV. This discrepancy was likely attributable to differences in the cancer type and gene panels (Fig. 1f). Restricting the analysis to shared regions in FFPE serial sections revealed comparable sensitivity (Supplementary Fig. 4c). When focusing on the common genes shared by the two iST platforms, Xenium 5K detected higher numbers of transcripts and genes per bin compared to CosMx 6K (Supplementary Fig. 4d).

For sST platforms, we evaluated quality metrics related to read alignment. Stereo-seq v1.3 exhibited a higher proportion of reads passing Unique Molecular Identifier (UMI) quality control compared to Visium HD FFPE and scRNA-seq, indicating improved retention of valid transcript information (Supplementary Fig. 4e). However, it also showed an elevated proportion of reads with invalid spatial barcodes, indicating greater loss of spatial information (Supplementary Fig. 4e). In addition, Stereo-seq v1.3 had a higher proportion of reads mapped to intergenic regions and multiple genomic loci compared to scRNA-seq (Supplementary Fig. 4f). To account for differences in sequencing

depth, we performed read downsampling and calculated the mean sequencing saturation across the ten selected ROIs from HCC and OV samples. Visium HD FFPE showed lower sequencing saturation for human transcripts at comparable sequencing depths (Fig. 1g).

### Evaluation of transcript background noise and diffusion control

Accurate transcript identification is essential for uncovering the underlying biological mechanisms using ST platforms. In iST platforms, negative probes and codes are used to assess nonspecific binding and fluorescence detection errors. For both iST platforms, negative control signals were evaluated alongside probes targeting the human transcriptome using 8 × 8 µm bins (Fig. 2a). Overall, CosMx 6K detected a higher total number of transcripts but exhibited reduced spatial variation and elevated negative control signals compared to Xenium 5K (Fig. 2b and Supplementary Fig. 5a, b). Spatial autocorrelation analysis using Moran's I revealed stronger aggregation of negative control signals in CosMx 6K (Fig. 2b and Supplementary Fig. 5a, b), indicating higher background interference. After normalization by total signal counts, Xenium 5K showed a lower proportion of negative control signals (Fig. 2c). Moreover, negative control signals in the necrotic regions of OV samples were markedly reduced in Xenium 5K, reflecting lower background noise (Supplementary Fig. 5c, d). Notably, across both platforms, negative probes consistently exhibited stronger signals than negative codes, indicating that nonspecific probe binding is the primary source of background noise in iST platforms (Fig. 2b, c, and Supplementary Fig. 5a–d). Across a range of quality control thresholds, Xenium 5K consistently maintained a lower proportion of negative control signals, highlighting its advantage in minimizing background noise (Supplementary Fig. 5e).

Transcript diffusion beyond tissue boundaries was observed in both Stereo-seq v1.3 and Visium HD FFPE (Fig. 2d and Supplementary Fig. 5f). To evaluate the diffusion, we measured transcript abundance in 8 µm bins located outside the tissue and calculated their distances from the tissue boundary. To reduce potential bias from differences in chip size, we restricted the maximum diffusion distance analyzed (**Methods**). Quantification of diffusion distance and transcript abundance revealed more effective diffusion control in Visium HD FFPE (Fig. 2e and Supplementary Fig. 5g, h). To account for differences in sequencing depth, we normalized the mean transcript counts of extratissue bins by those within the tissue. This analysis revealed substantially greater transcript diffusion in Stereo-seq v1.3 (Fig. 2f).

### Evaluation of transcript identification and localization accuracy

We utilized CODEX to profile proteins on tissue sections adjacent to each ST section, providing a high-resolution reference for evaluating transcript localization accuracy. To enable cross-modality comparisons, CODEX data were spatially registered to the corresponding ST datasets (Supplementary Fig. 6, **Methods**). We first assessed local concordance between ST and CODEX data in representative tissue areas. Tertiary lymphoid structures (TLS), aggregates of T and B cells that support both humoral and cellular immunity, play a crucial role in anti-tumor responses[37]. In COAD, we identified TLS-like structures and examined the spatial distribution of corresponding transcripts. Visium HD FFPE and Xenium 5K demonstrated strong spatial concordance with CODEX for B cell, CD4+ T cell, and CD8+ T cell markers (Supplementary Fig. 7a). In HCC, we leveraged the liver's intricate vascular architecture to evaluate the localization of *CD34* (endothelial cell marker) near vascular structures. Xenium 5K showed high levels of *CD34* transcripts along vascular edges (Supplementary Fig. 7b). In OV, we identified macrophage-rich regions and assessed *CD68* transcript localization, with all platforms detecting substantial *CD68* expression (Supplementary Fig. 7c). To evaluate global concordance, we annotated CODEX-derived cell types (Supplementary Fig. 7d, **Methods**) and compared the spatial patterns of diverse cell types with marker gene expression in adjacent ST sections. Visium HD FFPE and Xenium 5K

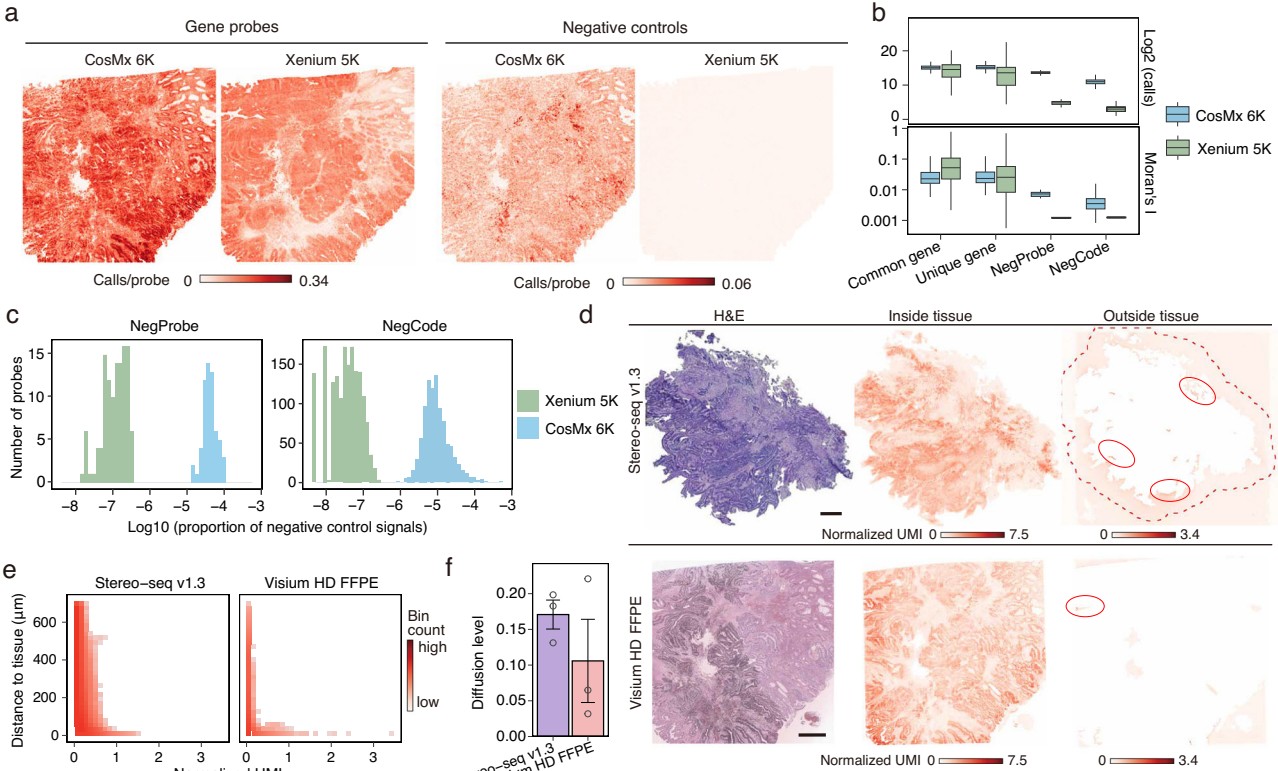

**Fig. 2 | Evaluation of false positives. a** Spatial distribution of gene calls (left) and negative control calls (right) in COAD samples profiled by CosMx 6K and Xenium 5K. Color intensity indicates mean call count per probe in each $8 \times 8\,\mu m$ bin. **b** Total call counts and Moran's I for common genes (2552), platform-specific genes (3623 for CosMx 6K and 2449 for Xenium 5K), negative probes (NegProbe, 20 for CosMx 6K and 40 for Xenium 5K), and negative codes (NegCode, 324 for CosMx 6K and 609 for Xenium 5K) detected by CosMx 6K and Xenium 5K across the shared regions shown in (**a**). Each data point represents one target. Center lines indicate the median value, and lower and upper hinges represent the 25th and 75th percentiles, respectively. The whiskers denote 1.5× the interquartile range. **c** Histogram showing the $\log_{10}$-transformed signal proportions of individual negative probes

and codes, pooled across COAD, HCC, and OV tissues. **d** H&E staining and transcript distribution inside and outside COAD tissue regions. Red dashed line outlines the Stereo-seq v1.3 region used for diffusion analysis; solid lines mark extra-tissue regions with high transcript levels. Color intensity indicates mean-normalized transcript count of each $8 \times 8\,\mu m$ bin. Scale bars, 1 mm. **e** Evaluation of transcript diffusion in COAD. The x-axis and y-axis represent the mean-normalized transcript counts and the distance to tissue edge for bins outside the tissue, respectively. Color intensity indicates the number of bins. **f** Ratio of the mean transcript count in extra-tissue bins to that in intra-tissue bins. Hollow circles indicate the ratio calculated for each of the three cancer types ($n = 3$). Data are presented as mean values +/− SEM. Source data are provided as a Source Data file.

showed higher concordance with CODEX than Stereo-seq v1.3 and CosMx 6K, respectively (Supplementary Fig. 7e). For epithelial cells, which are highly abundant in both COAD and OV, inter-platform differences were relatively minor (Supplementary Fig. 7f).

To address the inherent sparsity of ST data, we evaluated signature-level correlations in addition to individual marker genes. Differentially expressed genes for major cell types were identified from paired scRNA-seq data (Supplementary Fig. 8) and used to construct cell-type signatures for evaluating spatial concordance with CODEX. Within selected TLS-like ROIs, these cell-type signatures showed stronger spatial concordance with CODEX than individual markers (Fig. 3a and Supplementary Fig. 7a). Global concordance analysis indicated higher consistency for Visium HD FFPE and Xenium 5K over Stereo-seq v1.3 and CosMx 6K, respectively (Fig. 3b). As observed previously, the differences for epithelial cells among platforms remained minimal (Fig. 3c).

### Evaluation of the single-cell segmentation
Accurate cell segmentation is essential for ST platforms with subcellular resolution, as it significantly influences downstream analyses. We first compared automated cell segmentation results across platforms using four key morphological metrics: cell size, solidity (with higher values indicating greater convexity), aspect ratio (length-to-width ratio), and circularity (where 1 represents a perfect circle).

CosMx 6K and Xenium 5K performed cell segmentation based on multi-channel staining images that included nuclear, membrane, and cytoplasmic markers, while Stereo-seq v1.3 estimated cell boundaries by expanding nuclear masks by 5 μm. Visium HD FFPE was excluded from this analysis due to the absence of an official cell segmentation algorithm. Overall, CosMx 6K produced larger, more convex, and more circular cells, suggestive of more regular cell shapes (Supplementary Fig. 9a–d). To assess segmentation accuracy, we manually annotated nuclear boundaries for 72,405 cells across five regions (500 × 500 μm each) per dataset as ground truth (Supplementary Fig. 9e). The automatic cell segmentation results from CosMx 6K and Xenium 5K closely matched the number of cells manually identified by the manual nuclear segmentation within the same field of view, indicating their segmentation accuracy and reliability (Fig. 4a, b and Supplementary Fig. 9f, g). In contrast, Stereo-seq v1.3 exhibited reduced segmentation accuracy, likely due to staining artifacts that led to the misclassification of non-cellular structures as cells (Fig. 4a, b and Supplementary Fig. 9f, g).

Following cell segmentation, we compared transcript and gene counts within segmented cells across platforms. CosMx 6K and Xenium 5K retained a higher proportion of transcripts within the cell boundaries (Supplementary Fig. 10a), indicating more effective transcript assignment. Compared to the ST platforms, scRNA-seq consistently detected more transcripts and genes per cell (Fig. 4c). Marker

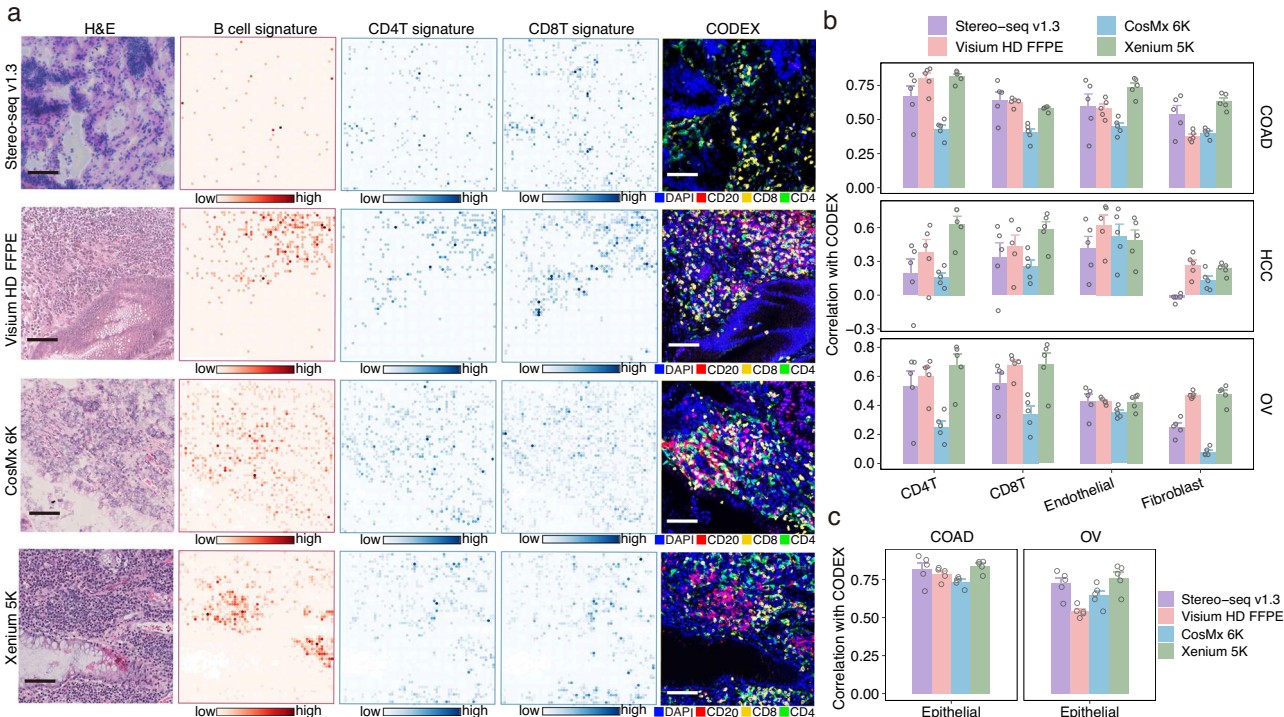

**Fig. 3 | Evaluation of transcript-protein correlation. a** Representative immune-enriched regions (each 500 × 500 μm) from COAD samples profiled using all four ST platforms. Left to right: H&E-stained histology, spatial distribution of transcriptomic signature scores for B cells, CD4+ T cell, and CD8+ T cells derived from ST data, and corresponding CODEX staining images (DAPI, CD20, CD8, CD4). For ST data, color intensity represents the corresponding signature score of each 8 × 8 μm bin. Scale bars, 100 μm. **b, c** Spatial correlation between CODEX-inferred cell counts and ST-derived signature scores for different cell types over the spatial grids. Panel **b** shows the correlations for immune and stromal signature scores, while panel **c** shows the correlations for epithelial signature scores. Pearson correlation coefficients are reported. Hollow circles indicate individual correlation values obtained under different grid sizes ($n = 5$). Data are presented as mean values +/− SEM. Source data are provided as a Source Data file.

gene expression levels also remained higher in scRNA-seq (Supplementary Fig. 10b). However, when restricting the analysis to a shared gene set (2522 genes), the iST platforms demonstrated comparable transcripts and genes per cell to scRNA-seq (Fig. 4c). Additionally, we performed whole-slide nuclear segmentation using StarDist[38] across the four ST platforms. Xenium 5K consistently identified nuclei with higher gene and transcript counts, both for all detected genes and for genes shared across platforms (Supplementary Fig. 10c).

Accurate cell segmentation is critical for resolving mutually exclusive transcripts into distinct cells, thereby improving cell type annotation. In COAD samples, ST data revealed substantial co-expression of the epithelial marker *EPCAM* and immune markers *CD3E* and *CD68* within the same cell (Fig. 4d). The co-expression levels observed in ST data were notably higher than those observed in scRNA-seq (Fig. 4d). A comparison of cell-level and bin-level co-expression further highlighted platform-specific differences in correctly assigning *EPCAM*, *CD68*, and *CD3E* to distinct cells (Supplementary Fig. 10d). To generalize these findings, we extended the analysis to 36 gene pairs derived from nine mutually exclusive lineage markers (Supplementary Data 6). scRNA-seq consistently exhibited lower co-expression than ST platforms (Fig. 4e). Among ST platforms, Xenium 5K showed the greatest reduction in artificial co-expression after segmentation (Fig. 4e), indicating better single-cell segmentation results.

### Evaluation of cell clustering and cell type annotation
Cell clustering is fundamental for characterizing cell heterogeneity and is critical for identifying novel cell types in both physiological and pathological contexts. To assess each platform's capacity to resolve cellular heterogeneity, we performed unsupervised clustering based on transcriptomic profiles (8 μm bins for Visium HD FFPE) (Fig. 5a).

Clustering performance, evaluated using the average silhouette score, showed that scRNA-seq provided the most effective separation of cell populations (Fig. 5b). Among the ST platforms, iST technologies achieved better resolution of transcriptomic differences between cell clusters, highlighting the advantage of their higher spatial resolution (Fig. 5b).

Cell type annotation is another key step in downstream analysis, critical for interpreting ST data. To evaluate the ability of different ST platforms to identify diverse cell types, we transferred annotations from matched scRNA-seq data to ST data using five annotation tools, including SELINA[39], Celltypist[40], Spoint[41], Tangram[42], and TACCO[43]. Among the ST platforms, CosMx 6K and Xenium 5K recovered a greater number of distinct cell types (Supplementary Fig. 11a). Notably, Xenium 5K exhibited a higher proportion of cells consistently annotated as the same cell type across five tools (Fig. 5c), suggesting better annotation robustness. Annotation consistency was further quantified by calculating entropy scores based on cell type assignments across tools. iST platforms showed higher entropy, indicating greater concordance in inferred cell type composition (Supplementary Fig. 11b). To obtain a consensus annotation, we integrated the outputs of the five tools using a majority vote approach (**Methods**). We then assessed concordance between ST and scRNA-seq data by computing pairwise gene expression correlations across annotated cell types. Stereo-seq v1.3 showed high agreement with scRNA-seq in COAD and OV samples, whereas Xenium 5K outperformed other platforms in HCC and OV (Supplementary Fig. 11c). Visualization of marker gene expression across annotated cell types revealed that Xenium 5K achieved the most distinct and biologically coherent expression patterns (Supplementary Fig. 11d), further supporting its annotation accuracy. Finally, we normalized gene expression using either the total expression of shared genes across platforms or the mean expression of housekeeping

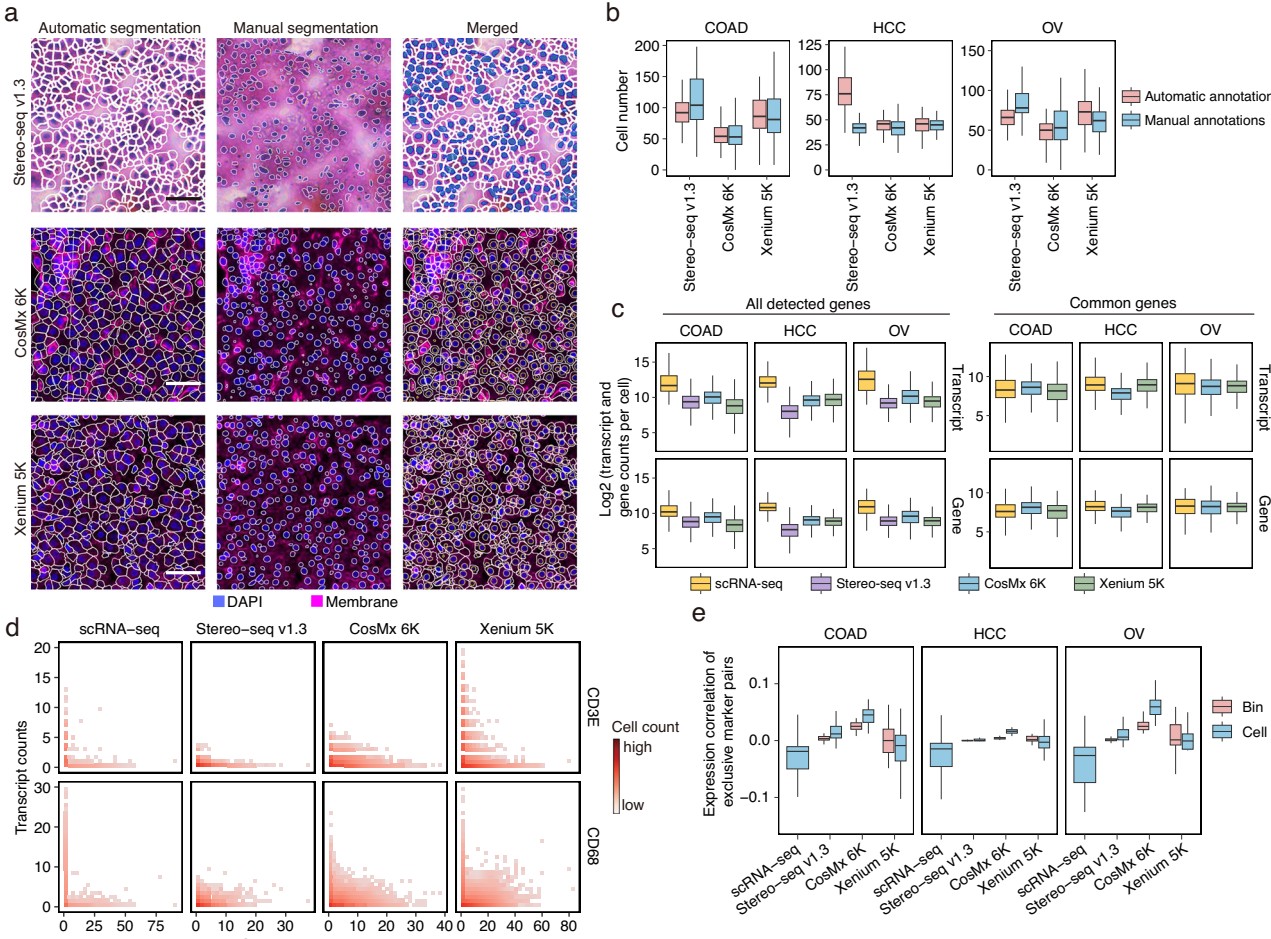

**Fig. 4 | Comparison of cell segmentation. a** Comparison of platform-derived automatic cell segmentation and manual nuclear segmentation across Stereo-seq v1.3, CosMx 6K, and Xenium 5K HCC sections. For each platform, a 250 × 250 μm region is shown. H&E staining for Stereo-seq v1.3 and multi-channel immuno-fluorescent staining for CosMx 6K and Xenium 5K are shown. Left column, automatically segmented cell boundaries. Middle column, manually segmented nuclear boundaries. Right column, overlay of automatic and manual segmentations, where white polygons denote automatic segmentations, and blue-filled masks in Stereo-seq v1.3 and yellow polygons in CosMx 6K and Xenium 5K indicate manual segmentations. Scale bars, 50 μm. **b** Number of automatically segmented cells and manually segmented nuclei per 100 × 100 μm bin across platforms (*n* = 125 bins per platform per cancer type). Each data point represents one bin. Center lines indicate the median value, and lower and upper hinges represent the 25th and 75th percentiles, respectively. The whiskers denote 1.5× the interquartile range. **c** Log$_2$-transformed transcript and gene counts per cell across platforms. For ST platforms,

the platform-derived automatic segmentations were used. Left: all detected genes included. Right: only retained genes shared across scRNA-seq, CosMx 6K, and Xenium 5K. Each data point represents one cell. Center lines indicate the median value, and lower and upper hinges represent the 25th and 75th percentiles, respectively. The whiskers denote 1.5× the interquartile range. **d** Joint density plots showing the expression of exclusive marker pairs within cells in COAD. Only cells with ≥1 transcript of either marker gene were included. Color intensity indicates the density of cells. **e** Expression correlation of 36 gene pairs expected to be exclusively expressed in distinct major lineages. Pearson correlation was computed across either cells or 8 × 8 μm bins. Each data point represents one marker pair (*n* = 36). Lower values indicate better separation of marker pairs. Center lines indicate the median value, and lower and upper hinges represent the 25th and 75th percentiles, respectively. The whiskers denote 1.5× the interquartile range. Source data are provided as a Source Data file.

genes. Xenium 5K most closely recapitulated scRNA-seq profiles for canonical cell type markers, including *CD34* (endothelial cells), *COL5A2* (fibroblasts), *CD68* (macrophages), and *CD8A* (T cells) (Supplementary Fig. 11e, f).

Advancements in iST platforms, particularly their capacity to capture large gene panels, enable more refined characterization of cell subtypes. To evaluate each platform's ability to recover T cell subtypes, we first established a reference based on matched scRNA-seq data (Supplementary Fig. 12a). Cell type annotations were then transferred to the ST datasets using five annotation tools. Among all platforms, CosMx 6K and Xenium 5K recovered the highest number of T cell subtypes (Supplementary Fig. 12b). Additionally, Xenium 5K exhibited a higher proportion of cells consistently annotated as the same subtype across tools, reflecting superior annotation reliability (Supplementary Fig. 12c).

We next evaluated the advantages of multimodal staining in cell segmentation and transcript assignment. In HCC samples, hepatocytes exhibited markedly irregular cellular morphologies relative to their nuclei. Moreover, distinct morphological differences were observed across cell types; in particular, hepatocytes displayed much less regular shapes compared to T cells (Supplementary Fig. 13a). Since Xenium 5K provided paired nuclear and cell segmentation, we used its results to evaluate the segmentations derived from different staining modalities. The irregular morphology of hepatocytes was effectively captured by the cell segmentation, as reflected by the lower solidity and circularity, highlighting the advantage of multichannel staining for improving cell segmentation accuracy (Supplementary Fig. 13b). Importantly, this improved segmentation enhanced transcript assignment accuracy: cell boundaries encompassed significantly more transcripts from identity-defining genes, such as cluster of

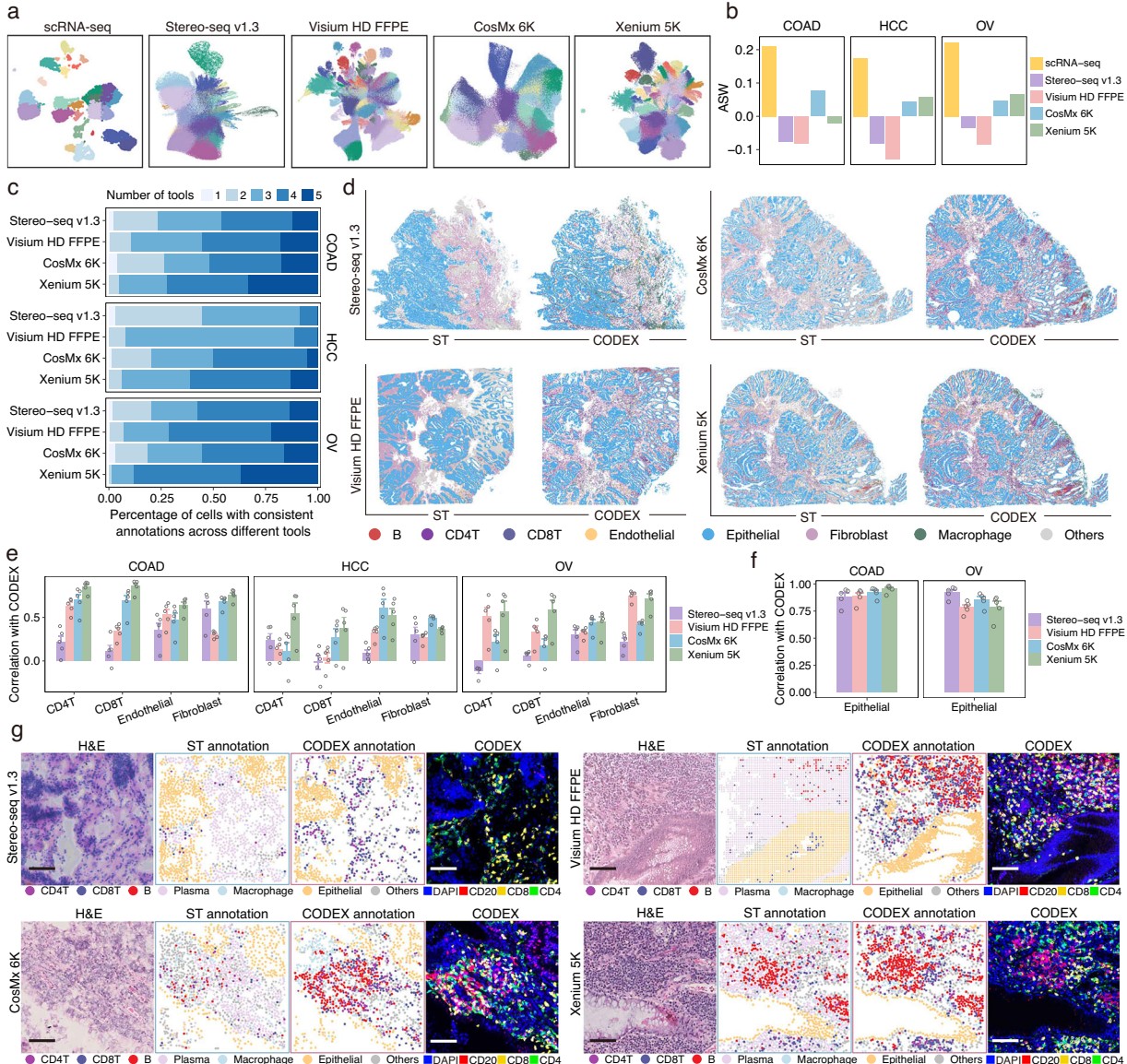

**Fig. 5 | Comparative analysis of cell clustering, cell type annotation, and spatial alignment with adjacent CODEX. a** Uniform Manifold Approximation and Projection (UMAP) of scRNA-seq and ST data for COAD samples. Each point represents a single cell (for scRNA-seq, Stereo-seq v1.3, CosMx 6K, and Xenium 5K) or an 8 × 8 µm bin (for Visium HD FFPE). Colors denote clusters identified by unsupervised clustering applied independently to each dataset based solely on transcriptomic profiles. **b** Average silhouette width (ASW) of unsupervised clustering results across platforms, with higher scores indicating better separation between distinct cell states. **c** Consistency of automated cell type annotations across five reference-based annotation tools. Bars represent the proportion of cells annotated as the same cell type by one to five tools. **d** Spatial distribution of annotated cell types in ST and CODEX data. Colors denote major cell types. Each ST platform is compared to its adjacent CODEX section. **e, f** Spatial correlation between CODEX-inferred and ST-inferred cell counts for different cell types over the spatial grids. Panel **e** shows the correlations for immune and stromal cells, while panel **f** shows the correlations for epithelial cells. Pearson correlation coefficients are reported. Hollow circles indicate individual correlation values obtained under different grid sizes (*n* = 5). Data are presented as mean values +/− SEM. **g** Representative immune-enriched regions (500 × 500 µm) from COAD sections. H&E staining, ST-derived annotations, CODEX-derived annotations, and multiplexed CODEX staining for CD20, CD8, and CD4 are shown. For ST data, each point represents a single cell (for Stereo-seq v1.3, CosMx 6K, and Xenium 5K) or an 8 × 8 µm bin (for Visium HD FFPE), colored by annotated cell type as shown in the legend. Scale bars, 100 µm. Source data are provided as a Source Data file.

differentiation markers (Supplementary Fig. 13c, d). Furthermore, multimodal cell segmentation enabled the identification of multinucleated cells, including neutrophils and hepatocytes, which were poorly resolved using nucleus-only segmentation approaches (Supplementary Fig. 13e−i). Together, these results underscore the added value of multimodal cell segmentation in improving transcript localization and delineating complex or multinucleated cells.

**Evaluation of ST cell type annotation accuracy against CODEX**
To evaluate the cell type annotation accuracy across ST platforms, we compared them with reference annotations derived from adjacent

CODEX-stained sections. In COAD samples, all ST platforms showed strong concordance with CODEX in capturing tissue architecture and cellular organization (Fig. 5d). To specifically assess immune cell detection, we quantified CD4+ T cells, CD8+ T cells, and macrophages in both ST and CODEX datasets (Supplementary Fig. 14a). Platforms demonstrating higher concordance with CODEX were considered more effective in identifying these immune cell types. Overall, iST platforms outperformed sST platforms in detecting lymphocytes, which are characterized by small cell sizes (Supplementary Fig. 14a). We also evaluated cell type annotation concordance by correlating cell-type-specific counts across spatial grids between ST and adjacent

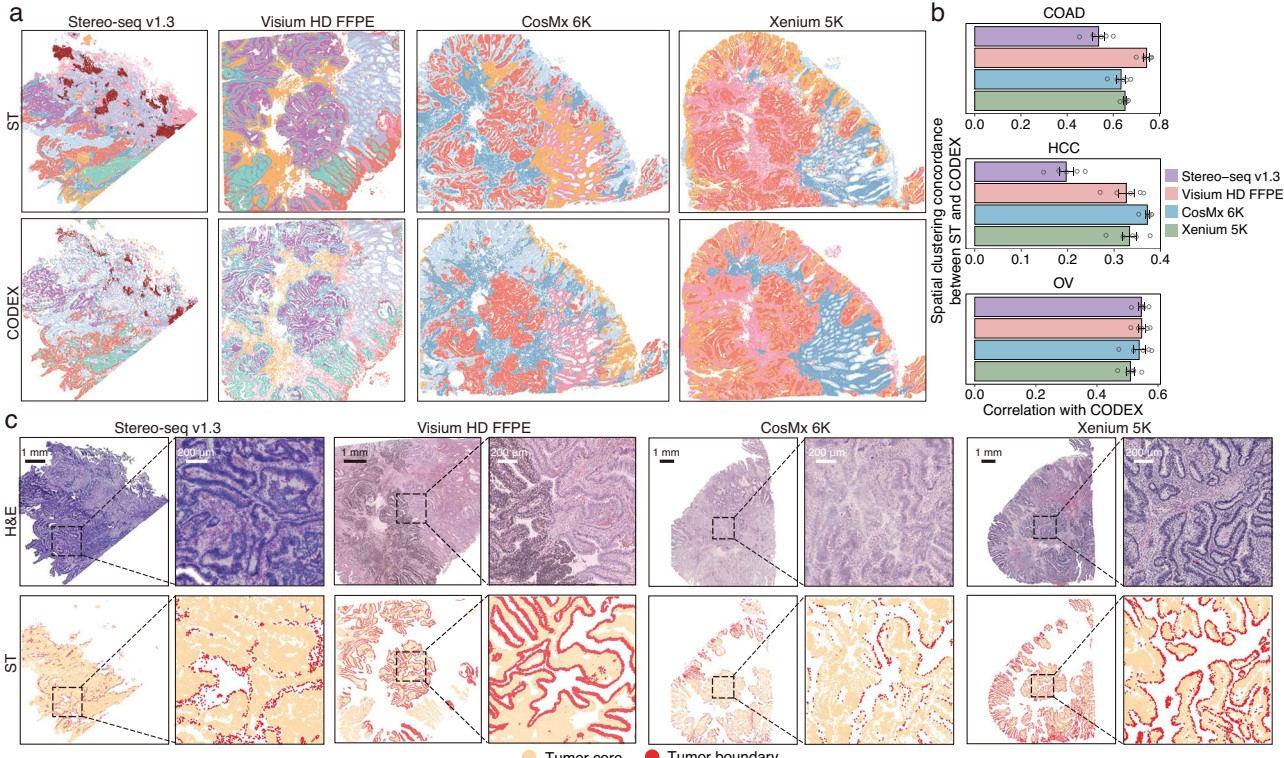

**Fig. 6 | Comparison of spatial clustering and malignant cell distributions.**
**a** Spatial clustering of COAD tissue sections profiled using four ST platforms and adjacent CODEX. Colors represent spatial clusters identified within each dataset. **b** Pearson correlation between cluster proportions in ST and CODEX data across spatial grids. Hollow circles indicate individual correlation values obtained under different grid sizes ($n = 5$). Data are presented as mean values +/− SEM. **c.** H&E staining and spatial localization of malignant cells at tumor boundary and core regions, as defined by unsupervised spatial clustering within each ST platform. Source data are provided as a Source Data file.

CODEX data. Xenium 5K exhibited the highest overall concordance, particularly for immune cells, while all platforms performed comparably in detecting epithelial cells, which are abundant and broadly distributed (Fig. 5e, f).

We further assessed the ability of different ST platforms to discriminate diverse immune cell populations in lymphocyte-enriched regions. Lymphocyte aggregates were identified in COAD sections for each ST platform (Fig. 5g), with marker genes associated with CD4⁺ T cells (*CD4*), CD8⁺ T cells (*CD8A*), B cells (*MS4A1*), plasma cells (*MZB1* and *CD38*) effectively captured (Supplementary Fig. 7a and Supplementary Fig. 14b, c). Xenium 5K demonstrated high efficiency and accuracy in identifying CD4⁺ T cells, CD8⁺ T cells, and B cells (Fig. 5g). Although Visium HD FFPE exhibited high sensitivity and specificity for markers such as *CD4*, *CD8A*, and *MS4A1* (Supplementary Fig. 7a), it demonstrated limited ability to distinguish CD4⁺ T cells, CD8⁺ T cells, and B cells from plasma cells (Fig. 5g and Supplementary Fig. 14b, c). This discrepancy is likely due to bin-level analyses resulting in mixed transcripts from neighboring cell types. Beyond COAD, we evaluated endothelial cell distribution in HCC. Both CosMx 6K and Xenium 5K successfully identified endothelial cells distributed along blood vessels, in agreement with anatomical expectations (Supplementary Fig. 14d). In OV, macrophage-enriched regions were detected by all ST platforms, demonstrating their robust capacity to identify macrophages (Supplementary Fig. 14e).

**Evaluation of spatial clustering and spatial pathway enrichment**
Spatial clustering plays a key role in identifying functional cellular aggregates within tissues. To evaluate the extent to which ST data can recapitulate spatial patterns observed in CODEX, we performed spatial clustering on both datasets using CellCharter[44] (Fig. 6a and

Supplementary Fig. 15a, b, **Methods**). Overall, comparable clustering concordance with CODEX was observed across ST platforms, with the exception of Stereo-seq v1.3 in HCC, which exhibited low concordance (Fig. 6b). This suggests that inter-platform differences in capturing large-scale tissue organization were relatively minor, although platform-specific limitations may affect performance in certain tissues. Spatial clustering also enabled the identification of spatially distinct cellular subtypes, potentially reflecting transcriptomic variations associated with tissue architecture. All ST platforms successfully identified malignant cells localized to either the tumor core or boundary, with Visium HD FFPE and Xenium 5K delineating more continuous tumor margins (Fig. 6c and Supplementary Fig. 15c). Additionally, spatial clustering revealed distinct subsets of CD8⁺ T cells, those infiltrating the tumor versus those positioned at the periphery, based on their spatial co-localization with cancer cells (Supplementary Fig. 15d). This highlights the utility of spatial clustering in characterizing immune cell localization and heterogeneity within the TME.

To evaluate each platform's ability to resolve biologically meaningful pathways, we examined the enrichment of functional pathways across matched tissue regions. Three region types, including immune cell-infiltrated areas, tumor regions, and normal epithelial regions, were selected from anatomically aligned positions on consecutive sections processed with Visium HD FFPE, CosMx 6K, and Xenium 5K (Supplementary Fig. 16a). Given that reduced sensitivity and specificity may limit the detection of region-specific gene expression, we first examined differentially expressed genes (DEGs) between immune-infiltrated and tumor regions, and between tumor and normal regions. Xenium 5K identified the highest number and proportion of DEGs across the three platforms, whereas CosMx 6K identified the fewest (Supplementary Fig. 16b–e). While some DEGs overlapped across

platforms, many were platform-specific (Supplementary Fig. 16d, e), highlighting differences in gene detection sensitivity and coverage.

We next performed Gene Ontology (GO) pathway enrichment analysis using these DEGs. In immune-infiltrated regions, Xenium 5K uniquely prioritized immune-related pathways, including T cell activation and leukocyte-mediated cytotoxicity, whereas Visium HD FFPE and CosMx 6K more frequently enriched for pathways related to extracellular matrix organization (Supplementary Fig. 16f). In tumor regions, Visium HD FFPE and Xenium 5K identified pathways associated with cell division and proliferation, while CosMx 6K enriched for pathways associated with metabolisms (Supplementary Fig. 16g). Across all comparisons, Xenium 5K detected the largest number of significantly enriched pathways (Supplementary Fig. 16h, i). Although some pathways were shared across platforms, each platform also revealed unique enrichments (Supplementary Fig. 16j, k), underscoring their differing sensitivities and gene coverage profiles.

## Discussion

In this study, we performed a comprehensive benchmarking of four cutting-edge ST platforms—Stereo-seq v1.3, Visium HD FFPE, CosMx 6K, and Xenium 5K—using treatment-naïve human tumor tissues. Compared to previous benchmarking efforts that primarily relied on histologically homogeneous tissues, our study focused on tumor samples characterized by high cellular heterogeneity, complex tissue architecture, and spatially irregular gene expression, thereby offering a more stringent and biologically relevant context. To ensure rigorous evaluation, we integrated orthogonal references, including high-resolution protein profiling via CODEX and matched scRNA-seq data. Beyond standard technical metrics such as sensitivity and specificity, we systematically evaluated downstream analytical capabilities, including cell type annotation, spatial clustering, and pathway-level enrichment, thereby establishing a comprehensive, multidimensional framework for comparing ST platform performance.

At the technical level, both sST and iST platforms demonstrated distinct strengths and limitations. Among the sST platforms, Visium HD FFPE outperformed Stereo-seq v1.3 in transcript detection sensitivity and specificity, likely due to its targeted transcript capture strategy and reduced transcript diffusion. Within the iST platforms, Xenium 5K exhibited stronger gene-wise correlations with scRNA-seq and more effective background signal control than CosMx 6K, which showed elevated signals for low-abundance genes. Notably, both Xenium 5K and Visium HD FFPE exhibited higher spatial concordance with CODEX-based protein maps, underscoring their spatial accuracy. Beyond transcript-level assessments, we also benchmarked cell segmentation performance. Xenium 5K achieved superior segmentation accuracy, minimizing transcript spillover between adjacent cells and enabling better resolution of single-cell boundaries. Importantly, it also captured more irregular cell morphologies, reflecting the well-established morphological heterogeneity among different cell types.

In downstream analyses, Xenium 5K demonstrated the highest accuracy in cell type annotation across all platforms, as evidenced by several key metrics. It achieved the greatest annotation consistency across multiple computational tools, exhibited cleaner marker gene expression, and showed the strongest concordance with adjacent CODEX protein data. Furthermore, Xenium 5K effectively delineated fine-grained spatial organization of specific cell types, such as immune cells within TLS-like regions and endothelial cells lining vascular structures. The use of multi-channel imaging enhanced annotation accuracy by enabling more precise cell boundary delineation and identification of multinucleated cells, a feature often missed by nucleus-only segmentation strategies. This multimodal segmentation approach improved transcript assignment to the correct cells, thereby providing more accurate cellular transcriptomic profiles. In contrast, Visium HD FFPE, while demonstrating high transcription detection sensitivity and specificity, showed reduced cell type annotation due to

the absence of reliable single-cell segmentation. For spatial clustering, all platforms successfully recapitulated large-scale tissue architecture. However, only Visium HD FFPE and Xenium 5K accurately delineated continuous tumor boundaries. In regional pathway enrichment analyses, Xenium 5K identified the highest number of differentially expressed genes and pathways due to its higher sensitivity and specificity.

Our findings provide practical guidance for selecting ST platforms based on study objectives. For single-cell level analyses, iST platforms are preferable due to their use of multichannel staining and full-cell segmentation, which enable more accurate cell boundary delineation and reduce transcript spillover. Their targeted capture strategies also enhance the sensitivity and specificity of marker gene detection, supporting cell state inference. Among the iST platforms, CosMx 6K was more susceptible to background noise, which may lead to false detection of low-abundance genes and reduce annotation fidelity. In contrast, for spatial analyses focused on tissue-region level patterns, sST platforms may be more suitable due to their broader gene coverage, which improves sensitivity for pathway-level enrichment analyses. Lastly, for applications involving host–microbe interactions, Stereo-seq v1.3 holds unique advantages due to its unbiased poly(A)-based capture, which allows for the detection of both human and non-human transcripts.

Looking ahead, each platform class faces distinct technical challenges. For iST technologies, expanding toward whole-transcriptome coverage while maintaining high specificity and detection efficiency is a critical frontier. However, optical crowding remains a limiting factor, as larger gene panels may reduce detection efficiency due to signal overlap[45]. Current strategies, such as reducing the number of probes per gene and increasing imaging cycles, show promise but require further innovations in probe design, codebook optimization, and the elimination of background fluorescence to maintain specificity. For sST platforms, improvements in spatial resolution, control of transcript diffusion, and accurate segmentation are needed to limit transcript leakage from neighboring cells. These may be addressed through advanced chip engineering and integration of DNA-staining-based segmentation methods. Importantly, permeabilization protocols remain a key determinant of transcript diffusion. Optimizing reagent formulations and incubation times can significantly reduce leakage[30]. This consideration is especially critical for FFPE samples relative to FF samples, as FFPE embedding can compromise RNA integrity. The resulting RNA fragmentation increases the likelihood of transcript leakage during the permeabilization process. Additionally, protocols could be improved during fresh-frozen tissue preparation, as slow freezing of tissues can induce ice crystal formation and membrane disruption, exacerbating transcript diffusion artifacts. Beyond these platform-specific considerations, broader technological advances will shape the future of ST development as well. The development of three-dimensional spatial transcriptomics for thick tissue sections, relaxation of sample input constraints, progress in non-destructive spatial profiling, and integration with multiomic modalities will collectively expand the scope, accessibility, and impact of ST technologies in both research and clinical settings.

Our study has several limitations that should be taken into account. First, we focused on commercialized ST platforms with larger gene panels, excluding those with smaller panels or limited commercial availability, which may narrow the scope of our comparisons. Second, CODEX was performed on adjacent tissue sections instead of the original ST sections, which provided essential reference but also introduced morphological discrepancies that may impact direct comparisons across platforms. Additionally, because Stereo-seq v1.3 relies on fresh frozen sections, its comparison with FFPE tissues was inherently constrained by structural differences associated with sample preparation methods. Third, our alignment and segmentation analyses utilized commercial pipelines, representing standardized

workflows commonly used by researchers but potentially missing insights from custom computational optimizations. Lastly, as our study exclusively used freshly collected samples, the generalizability of our findings to archived or long-term stored samples remains unclear, underscoring the need for further validation in a broader range of sample types and conditions.

Despite these limitations, our benchmarking study provides a comprehensive and biologically relevant evaluation of high-resolution ST platforms within the context of complex tumor tissues. By leveraging uniform sample processing, orthogonal multimodal references, and a multidimensional assessment framework, we offer practical guidance for selecting ST platforms tailored to specific biological questions, ranging from single-cell characterization to spatial pathway analysis. Beyond informing platform selection, the curated datasets, reference annotations, and evaluation metrics established in this study constitute valuable resources for the development, benchmarking, and optimization of next-generation spatial technologies. To further facilitate data exploration and reuse, we developed the SPATCH portal (https://spatch.pku-genomics.org/), which enables interactive visualization of gene expression and annotated cell type distributions across platforms and samples. The portal also provides access to both raw and processed ST and proteomic datasets, transcript-level spatial coordinate files, high-resolution histological images, multiplexed immunofluorescence-stained morphology images, and segmentation masks derived from both platform-specific pipelines and expert-curated nuclear boundaries. Together, these publicly available resources offer a valuable foundation for advancing spatial transcriptomics modeling, methodological innovation, and biological discovery.

## Methods

### Human sample collection and preprocessing

This study was approved by the Research and Biomedical Ethical Committee of Peking University (IRB00001052-24061) and conducted following pertinent ethical regulations. All patients provided informed consent for collecting clinical information and tumor samples. All protocols adhered to the Interim Measures for the Administration of Human Genetic Resources, administered by the Ministry of Science and Technology of China. Participants received no compensation for their participation. Sex was self-reported by participants, and no sex- or gender-related factors were incorporated into the study design or data analysis. The tumor specimens were obtained from three patients at the Chinese PLA General Hospital and Peking University People's Hospital, with each individual presenting a distinct cancer diagnosis: COAD, HCC, and OV. Necrotic areas and regions adjacent to major blood vessels were excluded during collection. Each tissue was further evenly divided into three sections. The middle portion was submerged in the MACS® Tissue Storage Solution (Miltenyi #130-100-008) and further processed for scRNA-seq. One of the remaining sections was fixed in a 10% neutral formalin fixing solution (Solarbio #G2161) for 24 to 48 h before paraffin embedding. The other section was embedded in 4 °C OCT compound (Sakura #4583), quickly frozen on dry ice, and transferred to a −80 °C freezer for storage until further experimentation. The entire process was completed within 30 min to minimize RNA degradation. Serial sections of FFPE samples were prepared at Peking University and loaded onto platform-specific chips under the supervision of trained technicians for Visium HD FFPE, Xenium 5K, and CosMx 6K. Sections adjacent to all ST sections were reserved for subsequent CODEX profiling. The tissue samples were destroyed after the analysis.

### Stereo-seq v1.3 data generation

Stereo-seq v1.3 assay is compatible with OCT-embedded tissues and H&E staining. Sample preparation and sectioning followed the Guide for Fresh Frozen Samples on Stereo-seq Chip Slides (Document No.: STUM-SP001). RNA quality was evaluated to determine whether to proceed with the following experiments. Only samples with RIN values ≥ 6 were accepted for further procedures. Cryosections were cut at a thickness of 10 μm in a Leica CM1950 cryostat. H&E staining, in situ reverse transcription, amplification, library construction, and sequencing followed the User Manual of the Stereo-seq Transcriptomics Set v1.3 (STOmics, #201ST13114 or 211ST13114). Tissue sections were loaded onto the Stereo-seq chip (generated by BGI, China; with a maximum size of 1 × 1 cm) and fixed in pre-cooled methanol. H&E staining was performed prior to tissue permeabilization. RNA was released from the permeabilized tissue, captured by the DNA nanoball (DNB), and subsequently underwent in situ reverse transcription. Following reverse transcription, tissue sections were removed to release complementary DNA (cDNA), which was purified using the VAHTSTM DNA Clean Beads (VAZYME #N411-02) and Stereo-seq 16 Barcode Library Preparation Kit (STOmics, #101KL160 or 111KL160). For library construction, 100 ng of cDNA was utilized for fragmentation and amplification. PCR products were purified using the VAHTSTM DNA Clean Beads. Ultimately, the purified PCR products were used for DNB production, and the libraries were sequenced using the MGI DNBSEQ-T7 sequencer.

### Visium HD for FFPE data generation

Visium HD assay is compatible with FFPE-embedded tissues and H&E staining. RNA quality of FFPE samples was assessed by calculating the percentage of RNA fragments >200 nucleotides (DV200) extracted from tissue sections. DAPI and H&E staining were also used to assess tissue morphology before performing the Visium HD assay. Tissue sections were cut at a thickness of 5 μm following the Visium HD FFPE Tissue Preparation Handbook (CG000684, 10x Genomics), spread out in RNA enzyme-free water at 42 °C, and loaded onto the slides prepared in advance (Fisher Scientific #1255015). Subsequently, these slides were air-dried at room temperature for 30 min and baked at 42 °C for 3 h. The subsequent experiments were carried out after drying overnight at room temperature.

Tissue sections were subjected to deparaffinization, H&E staining, and imaging following the Visium HD FFPE Tissue Preparation Handbook (CG000684, 10x Genomics). Probe hybridization, probe ligation, Visium HD slide preparation, probe release, extension, library construction, and sequencing followed the Visium HD Spatial Gene Expression Reagent Kits User Guide (CG000685, 10x Genomics). The tissue sections were destained and decrosslinked after H&E staining. The human whole transcriptome probe panel, consisting of about three specific probes per target gene, was added to the tissue sections. After hybridization, the Probe Ligation Enzyme (PN-2000425, 10x Genomics) was added to establish connections between the probe pairs hybridized to RNA, resulting in the formation of ligation products. The subsequent release and capture of these probes within the 6.5 × 6.5 mm capture areas were facilitated by the Visium CytAssist instrument following the User Guide. Treatment with RNase Enzyme and Perm Enzyme detached the single-stranded ligation products from the tissue and directed them onto the Visium HD Slide for capture. These ligation products were then elongated by adding the Spatial Barcode, UMI, and partial Read1 primer. Subsequent elution and amplification of the ligation products prepared them for indexing through the sample index PCR. The final libraries were cleaned up by SPRIselect. Sequencing was performed on an Illumina NovaSeq 6000 to obtain paired-end reads.

### Xenium 5K data generation

Xenium 5K assay is compatible with FFPE-embedded tissues and H&E staining. RNA quality of the tissue block was assessed by calculating the percentage of RNA fragments > 200 nucleotides (DV200) extracted from tissue sections. DAPI and H&E staining were also used to assess tissue morphology before performing the Xenium 5K assay.

Tissue sections were cut at a thickness of 5 µm following the Xenium In Situ for FFPE-Tissue Preparation Guide (CG000578, 10x Genomics), spread out in RNA enzyme-free water at 42 °C, and attached to the Xenium slides (PN-3000941, 10x Genomics) within the sample area (with a maximum size of 10.45 × 22.45 mm) without overlapping with the surrounding fiducials. The slides were dried at room temperature for 30 min and baked at 42 °C for 3 h. The follow-up experiment was carried out after drying overnight at room temperature.

After drying overnight, the Xenium slides were subjected to deparaffinization and decrosslinking following the Xenium In Situ Protocol for FFPE-Deparaffinization and Decrosslinking (CG000580, 10x Genomics). Priming hybridization, RNase treatment & polishing, probe hybridization, probe ligation, amplification, cell segmentation staining, autofluorescence quenching, and nuclear staining followed the Xenium Prime In Situ Gene Expression with optional Cell Segmentation Staining (CG000760, 10x Genomics). The Xenium Cell Segmentation Staining Reagents (PN-1000661, 10x Genomics) were used for membrane, cytoplasm and nuclear staining. The assay was performed using the Xenium Prime 5K Human Pan Tissue & Pathways Panel (PN-1000724, 10x Genomics), which targets 5,001 individual human genes. After priming hybridization and RNase treatment & polishing steps, the Xenium slides were incubated with probes at 50 °C for 16-24 h for probe hybridization and then washed with PBS-T. Then the slides were subjected to probe ligation at 42 °C for 30 min, amplification enhancement at 4 °C for 2 h, and amplification at 30 °C for 1.5 h. Following additional washing procedures, the slides underwent cell segmentation staining, then treatment with an autofluorescence suppressor and nuclear staining. The slides were loaded onto Xenium Analyzer (PN-1000529, 10x Genomics) according to the Xenium Analyzer User Guide (CG000584, 10x Genomics) and run for about 90 h. The Xenium Onboard Analysis pipeline v.3.1.0 (10x Genomics) was run directly on the instrument for imaging processing, cell segmentation, image registration, decoding, deduplication, and secondary analysis. After that, the slides were washed to perform post-run H&E staining.

## CosMx 6K data generation

CosMx 6K assay is compatible with FFPE-embedded tissues and H&E staining. RNA quality of the tissue block was assessed by calculating the percentage of RNA fragments >200 nucleotides (DV200) extracted from tissue sections. DAPI and H&E staining were also used to assess tissue morphology before performing the CosMx 6K assay. Tissue sections were cut at a thickness of 5 µm following the CosMx SMI Manual Slide Preparation for RNA Assays (MAN-10184-02, NanoString Technologies), spread out in RNase-free water at 42 °C, and attached to the slides (CIITOTEST #188105) within the scan area (with a maximum size of 2.0 × 1.5 cm). The slides were dried at room temperature for 30 min and baked at 65 °C for 30 min. The follow-up experiment was carried out after drying overnight at room temperature.

Deparaffinization, target retrieval, protease digestion, blocking, hybridization, stringent washing, blocking, nuclear and segmentation markers staining, and imaging followed the CosMx SMI Manual Slide Preparation for RNA Assays (MAN-10184-02, NanoString Technologies). Human 6K Discovery Panel, 6K-plex, RNA (#121500041, NanoString Technologies) was used. The tissue sections were deparaffinized, subjected to target retrieval at 100 °C for 15 min, treated with protease for digestion at 40 °C for 30 min, incubated with applied fiducials for 5 min, post-fixed, blocked, and incubated with the human 6K Discovery Panel overnight. The slides were washed and blocked, followed by nuclear staining. Then the sections were incubated with Marker Stain Mix (PanCK, CD45) and Cell Segmentation Mix (CD298, B2M) using CosMx™ Human Universal Cell Segmentation Kit (RNA) (121500020, NanoString Technologies). The slides were washed again and loaded onto the CosMx SMI system (cat #101000, S/N: SMI_2307H0124) for UV bleaching, imaging acquisition, cycling

processing, and scanning according to the Instrument User Manual (MAN-10161-05, NanoString Technologies). The raw images were subsequently decoded using Atomx (v.1.3.2). Finally, the slides were washed to perform post-run H&E staining.

## CODEX data generation

Tissues embedded in FFPE were sliced, spread out in RNase-free water at 42 °C, and loaded onto the slides (Fisher Scientific #1255015) within the scan area (with a maximum size of 3.5 × 1.8 cm). The slides were dried at room temperature for 30 min and baked at 65 °C for 30 min. The follow-up experiment was carried out after baking overnight at 60 °C. A 16-plex commercial antibody panel was used to target 16 proteins. The sample preparation, tissue staining, and imaging followed the PhenoCycler-Fusion User Guide_2.2.0 (PD-000011 REV M, Akoya Biosciences). After overnight baking, the slides were subjected to deparaffinization and antigen retrieval using a sodium citrate solution for 20 min at 11.6PSI/110 °C. Subsequently, the slides were washed using a hydration buffer (P/N 7000017, Akoya Biosciences) and incubated in a staining buffer (P/N 7000017, Akoya Biosciences) at room temperature for 20 min. A mixture of antibodies, blocking solution, and staining solution was prepared. The slides were incubated with the staining mix at room temperature for 3 h. The antibody dilution ratio was determined based on pre-tests, along with the cycle information summarized in Supplementary Data 7. Following staining, the tissues were sequentially fixed with PFA, ice-cold methanol, and a final fixative solution. The slides were washed, loaded onto the PhenoCycler-Fusion instrument (PhenoCycler-Fusion 2.0), and imaged according to the instrument's instructions.

For the FF samples, cryosections were cut at a thickness of 10 µm in a Leica CM1950 cryostat, loaded onto the slides (Fisher Scientific #1255015) within the scan area (with a maximum size of 3.5 × 1.8 cm) and stored at −80 °C before the experiment. The sample preparation, tissue staining, and imaging followed the PhenoCycler-Fusion User Guide_2.2.0 (PD-000011 REV M, Akoya Biosciences). The slides were dried and warmed for 5 min at room temperature, fixed in acetone for 10 min, incubated with the hydration buffer (P/N 7000017, Akoya Biosciences), fixed with 1.6% PFA for 10 min, and finally balanced with staining buffer (P/N 7000017, Akoya Biosciences) at room temperature for 20 min. A mixture of antibodies, blocking solution, and staining solution was prepared. Then the slides were incubated with the staining mix at room temperature for 3 h. The antibody dilution ratio was consistent with that used for FFPE samples. Following staining, the tissues were sequentially fixed with PFA, ice-cold methanol, and a final fixative solution. The slides were washed, loaded onto the PhenoCycler-Fusion instrument (PhenoCycler-Fusion 2.0), and imaged according to the instrument's instructions.

## scRNA-seq data generation

Single-cell suspensions from primary human tumor tissue were generated using the Tumor Dissociation Kit (Miltenyi Biotec, #130-095-929). The proportion of viable cells exceeded 85% in all samples. The single-cell suspension was processed with the Chromium Single Cell 3' GEM, Library & Gel Bead Kit v3.1 (10x Genomics, PN-1000268) and loaded onto a Chromium Single Cell Chip (Chromium Single Cell G Chip Kit, 10x Genomics, PN-1000120) according to the manufacturer's instructions for co-encapsulation with barcoded Gel Beads. The captured cells were lysed, and the released RNA was barcoded through reverse transcription in individual single-cell gel beads in the emulsion (GEMS). In each droplet, cDNA was generated and amplified through reverse transcription on a T100 PCR Thermal Cycler (Bio-Rad) at 53 °C for 45 min, followed by 85 °C for 5 min and a hold at 4 °C. Then, cDNA concentration and quality were assessed using a Qubit Fluorometer (Thermo Scientific) and bioanalyzer 2100 (Agilent), respectively. scRNA-seq libraries were then constructed and sequenced on the Illumina platform according to the manufacturer's introduction.

## Collection of gene sets

Cell membrane proteins, which span or embed within the plasma membrane, facilitate communication between cells and the extracellular environment. Both experimental and computational approaches have been employed to identify and predict cell-surface membrane proteins. However, each method has inherent limitations, often resulting in incomplete coverage and false positives[46–48]. Among the various resources available, we selected the latest and most comprehensive database related to cancer research[49]. Ligands, receptors, cytokines, and transcription factors were also collected from previously published studies and databases[50–54].

## Data preprocessing

scRNA-seq data were processed with cellranger (v.7.0.0). Visium HD FFPE data were processed with spaceranger (v.3.0.0). Stereo-seq v1.3 data were processed with SAW (v.8.0). GRCh38 was used as the reference genome. For Xenium 5K, we retained the calls with Phred-scaled quality scores higher than 20. The calls from CosMx 6K underwent filtration based on the methodologies described in the previous study[24]. The morphology staining images of all field of views from CosMx 6K were stitched using napari-cosmx (https://github.com/Nanostring-Biostats/CosMx-Analysis-Scratch-Space). We performed tissue masking to remove the calls located outside the tissue using the Python package OpenCV (v.4.10.0) for both Xenium 5K and CosMx 6K. To enable a fair comparison, we binned the data from Stereo-seq v1.3, CosMx 6K, and Xenium 5K at a resolution of 8 μm, and used the 8 μm resolution output of Visium HD FFPE for basic metric evaluations.

## Gene-wise expression correlation between ST and scRNA-seq data

The total transcript count of each gene was $log_{10}$-transformed, and gene-wise correlations between ST and scRNA-seq expression profiles were computed. For CosMx 6K, transcript calls were ranked according to their probability of representing random signals. We then stepwise extracted the top percentages of high-quality calls—specifically at 50%, 62.5%, 75%, and 87.5%—and assessed the gene-wise expression correlations with scRNA-seq data.

## Sequencing reads downsampling for Stereo-seq v1.3 and Visium HD FFPE data

The --unmapped-fastq option was used in SAW to retain unmapped reads for Stereo-seq v1.3. We selected ten regions characterized by a high density of cancer cells in each dataset based on H&E staining. BAM files were filtered to isolate the reads with valid UMI information. The valid reads with spatial coordinates mapped to the ten regions were downsampled to fixed proportions (20%, 40%, 60%, and 80%) using the Python package pysam (v.0.22.1).

## Evaluation of diffusion control for Stereo-seq v1.3 and Visium HD FFPE data

To account for variability in sequencing depth, the total transcript count for each bin was normalized by the mean transcript count across all bins. Diffusion distance was defined as the Euclidean distance from each bin located outside the tissue boundary to its nearest neighboring bin within the tissue, computed using the NearestNeighbors function from the Python package scikit-learn (v.1.5.2). Given the larger chip size of Stereo-seq v1.3 and its potential for long-range transcript diffusion, we restricted our analysis to bins with diffusion distances shorter than the maximum observed in Visium HD FFPE, thereby ensuring comparability across platforms.

## Registration of images and alignment of spatial data

We manually annotated key landmarks on paired images and employed the SimpleITK library (v.2.4.0) to achieve accurate registration. Specifically, the Similarity2DTransform, SetMetricAsMattesMutualInformation,

and sitkLinear functions were utilized to perform automated adjustment after the initial transformation derived from paired landmarks. For the alignment of Visium HD FFPE, CosMx 6K, and Xenium 5K data, we set the grayscale H&E image of Visium HD FFPE as the fixed reference and registered the DAPI images of CosMx 6K and Xenium 5K to it. The derived transformations were subsequently applied to map the CosMx 6K and Xenium 5K data onto the coordinate system of the Visium HD FFPE data. For the alignment of ST data with adjacent CODEX data, the grayscale H&E images of Stereo-seq v1.3 and Visium HD FFPE were used as the fixed references. For CosMx 6K and Xenium 5K, the DAPI images were used as the fixed references. The fixed references were rescaled to match the resolution of CODEX, and the DAPI channel of CODEX was registered to these references. The derived transformations were subsequently applied to the remaining CODEX channels. To enable direct comparisons across FFPE samples, we used the Python package OpenCV to extract tissue masks of each ST data based on the paired staining images and intersected them to define the shared regions. A similar approach was used to extract overlapping regions between ST data and adjacent CODEX data.

## Annotation of scRNA-seq data

Genes detected in fewer than 10 cells were excluded from the analysis. Cells that did not fulfill the following criteria were removed: $1,000 \leq UMI \leq 25,000$, $500 \leq Gene \leq 5,000$, and percentage of mitochondrial genes $\leq 10\%$. Putative doublets were identified and removed using DoubletFinder[55] (v.2.0.3). A two-round clustering strategy was applied for cell type annotation using Seurat[56] (v.5.1.0). In the first round of clustering, the data were normalized and log-transformed to the same scale. A set of 2000 highly variable genes was identified, followed by scaling of the expression matrix. The top 30 principal components (PCs) were identified to build a nearest-neighbor graph. Clustering was performed using the shared nearest neighbor (SNN) modularity optimization algorithm. We annotated each cluster based on its expression of the following known markers: B cell, *CD79A*, *CD19*, and *MS4A1*; cDC1, *XCR1* and *CLEC9A*; cDC2, *CD1C* and *CLEC10A*; mregDC, *LAMP3* and *CCR7*; pDC, *LILRA4*; macrophage, *CD68*, *C1QC*, and *SPP1*; mast cell, *KIT* and *TPSAB1*; monocyte, *FCN1*; neutrophil, *CSF3R* and *AQP9*; endothelial, *VWF*, *CD34*, *CDH5*, and *PECAM1*; fibroblast, *ACTA2*, *COL1A2*, and *FAP*; SMC, *ACTA2* and *RGS5*; NK cell, *FCGR3A*, *GZMA*, and *NCAM1*; plasma cell, *SDC1* and *MZB1*; CD4$^+$ T cell, *CD4*, *CD3G*, *CD3D*, and *CD3E*; CD8$^+$ T cell, *CD8A*, *CD8B*, *CD3G*, *CD3D*, and *CD3E*; Tprolif, *MKI67*, *CD3G*, *CD3D*, and *CD3E*; epithelial, *EPCAM*; hepatocyte, *ALB*; kupffer cell, *CD5L*. The subtypes of T cells were annotated after a second round of clustering using a similar approach.

## Evaluation of segmentation results

Manual segmentation was conducted using the software Labelme (v.5.5.0), where nuclear outlines were drawn manually based on DAPI and H&E staining. Solidity was calculated as the ratio of the contour area to its convex hull area, with higher values indicating convexity and lower values suggesting concavity. Circularity measures how closely a shape approximates an ideal circle, with a value of 1 corresponding to a perfect circle and 0 indicating a more irregular shape. The aspect ratio is the ratio of the width to the height of the bounding box that encloses the contour, which describes the elongation of the shape, with values greater than 1 indicating horizontal elongation and values less than 1 indicating vertical elongation. All metrics were calculated using the Python library OpenCV.

## Evaluation of the clustering of ST data based on the transcriptomic profiles

Genes detected in fewer than 100 bins or cells were filtered out. Bin-level data demonstrated lower quality than cell-level data due to the retention of non-cellular regions that would be excluded after cell segmentation. To address this, we applied a more stringent cutoff to

filter out the low-quality bins. For Visium HD FFPE, bins with total counts below the 20th percentile of all bins were excluded. For cell-level data from Stereo-seq v1.3, CosMx 6K, and Xenium 5K, we filtered out low-quality cells with total counts below the 10th percentile of all cells. We further utilized the Python package scanpy (v.1.10.3) to perform clustering. Data were normalized and log-transformed to the same scale. The top 10% of genes with the highest variance were defined as highly variable genes. The top 30 principal components were computed to build neighborhood graphs. Data were embedded using Uniform Manifold Approximation and Projection (UMAP) for further dimensionality reduction and visualization. Clustering was performed using the Leiden algorithm with default resolution settings. The silhouette_score function from the Python package scikit-learn was used to assess the clustering quality. This score evaluates cluster separation by comparing intra-cluster and inter-cluster distances, with a value approaching 1 indicating better-defined clusters.

### Annotation of ST and CODEX data

The latest versions of SELINA[39] (v.0.1), Celltypist[40] (v.1.6.3), Spoint[41] (v.1.1.7), Tangram[42] (v.1.0.4), and TACCO[43] (v.0.4.0.post1) were used to transfer the annotations from scRNA-seq data to the filtered ST data. We utilized two metrics to evaluate the annotation consistency across different tools: (1) the proportion of cells that were consistently annotated as the same cell type by different tools; (2) the entropy of cell numbers detected by different tools for each cell type, which was calculated with the following formula:

$$p_i = \frac{N_i}{\sum_{i=1}^{n} N_i} \tag{1}$$

$$H = -\sum_{i=1}^{n} p_i \, log_2(p_i) \tag{2}$$

where the number of cells detected by the $i$-th tool was denoted with $N_i$ and its ratio over all tools was denoted as $p_i$. Each cell was assigned a final cell type based on majority voting across annotation tools. For cells with inconsistent annotations across five tools, the label from the method showing the highest overall concordance with other tools was used to resolve conflicts.

For the CODEX data, we first performed nuclear segmentation using StarDist[38] (v.0.5.0) in QuPath[57] (v.0.5.1). Cell boundaries were defined by expanding the nuclear boundaries by 5 μm. CODEX data exhibited prominent non-specific binding signals in tumor regions and background signals across entire sections, which could bias cell annotation if based solely on the average signal intensity. To address this challenge, we manually labeled hundreds of positive cells for each marker and trained a k-nearest neighbor (KNN) classifier in QuPath. For membrane markers, cells with fluorescent signals surrounding the nuclei were defined as truly positive cells. For transcription factors, cells with fluorescent signals confined exclusively to the nuclei were defined as truly positive cells. Cells lacking any marker signal were categorized as negative cases. We trained the classifier using various signal statistics, including mean, median, minimum, maximum, and standard deviation for signals in the nucleus, cytoplasm, membrane, and the entire cell. This classifier was then applied to annotate the remaining cells across the entire section.

### Correlation between ST and CODEX data

We evaluated the concordance between CODEX-based cell annotations and ST-derived features including: (1) individual marker gene expression, (2) cell-type-specific gene signatures, and (3) cell type abundance. To construct gene signatures, we selected the top 15 differentially expressed genes for each cell type from matched scRNA-seq data, and retained those present in at least two tissue types. To mitigate platform-specific gene panel differences, we intersected these genes with common genes shared by all four ST platforms. The shared regions of ST and CODEX data were binned at multiple spatial resolutions (100, 200, 300, 400, and 500 μm). For each spatial bin, we quantified ST-derived features at all three levels, alongside corresponding cell counts inferred from CODEX. Pearson correlation coefficients were then computed across all spatial bins to assess the spatial concordance between the ST-derived features and CODEX annotations. Smooth muscle cells (SMCs) and fibroblasts identified by the ST platforms were both categorized as fibroblasts in alignment with CODEX data, as they all express ACTA2, which encodes α-SMA—the marker used to annotate fibroblasts in the CODEX data.

### Spatial clustering of ST and CODEX data

CellCharter[44] was used to perform spatial clustering on both ST and CODEX data. CODEX enables accurate measurement of marker protein distribution across major cell types, providing a reliable reference for tissue architecture. Given this, we leveraged the optimal number of clusters derived from CODEX data to guide the clustering of ST data. For HCC, CD34, CD4, FOXP3, and HLA-A channels were excluded from CODEX clustering due to quality concerns. To investigate the clustering concordance between CODEX and ST data, we computed cell proportions within spatial grids (100, 200, 300, 400, and 500 μm) for each cluster and assessed their correlations across all grids. For each CODEX cluster, we identified the best-matching ST cluster based on the maximal correlation, and the final metric was defined as the average correlation across all matched cluster pairs.

### Spatial pathway enrichment

To assess pathway-level differences across spatial regions, differentially expressed genes (DEGs) were identified using the scanpy Python package. Comparisons were made between immune-infiltrated versus tumor regions and tumor versus normal epithelial regions. Genes with adjusted $p$-value ≤ 0.05 and fold change ≥ 2 were defined as DEGs and subsequently subjected to GO enrichment analysis (R package clusterProfiler[58], v.4.6.2). Pathways with adjusted p-value ≤ 0.05 were retained for downstream comparisons across platforms.

### Statistics & reproducibility

No statistical method was used to predetermine sample size. No data were excluded from the analyses. The experiments were not randomized. The Investigators were not blinded to allocation during experiments and outcome assessment. All H&E staining was performed by the respective commercial providers corresponding to each spatial transcriptomics platform. CODEX experiments were carried out by the Optical Imaging Core Facility, National Center for Protein Sciences at Peking University. Due to the limited availability of samples and experimental kits, each experiment was performed once and no independent replicates were generated.

### Reporting summary

Further information on research design is available in the Nature Portfolio Reporting Summary linked to this article.

## Data availability

The raw sequencing data have been deposited in the Genome Sequence Archive at the National Genomics Data Center under accession number HRA011129. The image data have been deposited in BioImage Archive under accession number S-BIAD1900. Both raw and processed data are publicly accessible on the SPATCH website at http://spatch.pku-genomics.org/. Beyond data download, this web server offers tools for data visualization and exploration, enabling users to interactively analyze the datasets. Source data are provided with this paper.

## Code availability

The code utilized for data processing and analysis in this study is publicly available on GitHub (https://github.com/zenglab-pku/SPATCH)[59].

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

## Acknowledgements

This work was supported by the National Natural Science Foundation of China (92374116, 32470664, T2321001 to Zexian Zeng; 82404094 to Zhe Zhang), Beijing Natural Science Foundation (L248043, Zexian Zeng), Noncommunicable Chronic Diseases-National Science and Technology Major Project (2024ZD0520600, Zexian Zeng), Sichuan Science and Technology Program (2024YFFK0064, Zexian Zeng), Beijing Advanced Center of Cellular Homeostasis and Aging-Related Diseases (Zexian Zeng), and Peking-Tsinghua Center for Life Sciences (Zexian Zeng). We thank the Optical Imaging Core Facility, National Center for Protein Sciences at Peking University, especially Dr. Chunyan Shan and Ms Yan Luo, for technical assistance with the CODEX experiment. Part of the analysis was performed on the High-Performance Computing Platform of the Center for Life Sciences at Peking University.

## Author contributions

Z.Z. (Zexian Zeng), Z.Z. (Zhe Zhang), and Y.M. conceived the study and designed experiments. P.R. performed the bioinformatics analysis with help from Y.W., C.L., and S.W. R.Z., Y.L., and X.L. collected and processed the tumor samples. P.Z. developed the SPATCH website. Z.Z. (Zongxu Zhang), Y.Z. (Yanping Zhao), Y.H. and H.Z. helped with the study design and analysis of ST data. Z.G., X.Z., Y.Z. (Yahui Zhao), and Z.L. provided tumor samples and clinical annotations. P.R., R.Z., Z.Z.(Zhe Zhang) and Z.Z. (Zexian Zeng) wrote and revised the manuscript with the help of other authors.

## Competing interests

The authors declare no competing interests.
