## [Transparent Peer Review file · Nature Communications]

Systematic Benchmarking of High-Throughput Subcellular Spatial Transcriptomics Platforms Across Human Tumors

Corresponding Author: Dr Zexian Zeng

Version 0:

Reviewer comments:

Reviewer #1

(Remarks to the Author)

The study aims to compare high-throughput spatial transcriptomics (ST) platforms, comparing sequencing-based (sST) and imaging-based (iST) approaches. While the fundamental trade-offs between these methodologies — iST platforms, which offer superior spatial accuracy but limited transcriptome-wide profiling, and sST platforms, which offer broader coverage at the expense of spatial precision — are well recognised, this study helps to refine existing knowledge through more rigorous comparisons. In particular, it minimises batch effects through standardised conditions, includes multi-omics validation by single-cell RNA sequencing (scRNA-seq) and CODEX proteomics, and provides a publicly available dataset (SPATCH). While these efforts improve methodological rigour, the study does not fundamentally redefine our understanding of the performance of ST platforms. The main conclusions regarding the trade-offs between platforms align closely with previous studies, which raises the question of whether the primary value of the study lies in new insights or simply a higher quality benchmark dataset. The following points highlight areas where additional justifications, methodological refinements and clearer conclusions could improve the manuscript.

1. The different spatial transcriptomics platforms in this study used either fresh-frozen (FF) or formalin-fixed paraffin-embedded (FFPE) tissue sections. Given that all platforms support FF, can the authors justify why different fixation methods were chosen for the different platforms? This choice introduces additional variability as FF and FFPE samples undergo different preservation processes that may affect RNA integrity and gene expression profiles. In addition, the need to cut different tissue blocks introduces further variability and may affect the efficiency of direct cross-platform comparisons. Given these factors, how do you account for and mitigate potential biases introduced by these methodological differences? If not, how might this affect direct platform comparisons?

2. The study finds that different spatial transcriptomics platforms perform better in certain tumour types. What factors do the authors believe have contributed to these platform- and sample-dependent variations? Also, do the authors believe that certain platforms are inherently better suited to certain tumour microenvironments, or do these differences arise primarily from experimental conditions and sample processing?

3. The Xenium In Situ Cell Segmentation Kit (10x Genomics) and NanoString's CosMx™ Human Universal Cell Segmentation Kit provide advanced multimodal segmentation by staining intracellular RNA, proteins, cell borders and nuclei for improved cell segmentation. Given that segmentation accuracy critically affects downstream spatial transcriptome analyses, can the authors explain why they did not include these advanced segmentation kits in their assessment? Do they anticipate that the use of such multimodal approaches would increase the accuracy of transcript localisation within cells and improve the overall representation of cell shape, particularly for imaging-based ST platforms? Furthermore, have alternative segmentation refinements been attempted post hoc (e.g., deep learning-based segmentation) to mitigate challenges in complex cell types? Additionally, how do you think these methods would perform in multinucleated cells (e.g., skeletal muscle fibres, syncytial trophoblasts, osteoclasts), where standard nuclear-based segmentation may be insufficient to capture their true morphology and transcript distribution?

4. The study evaluates marker gene detection and molecular capture efficiency on different platforms, but it is unclear whether relative gene expression within the same cell types on different platforms was explicitly compared. Given that different ST platforms have different efficiencies in capturing transcripts, did the authors attempt to normalise gene expression across platforms to allow direct comparisons between specific cell types, such as endothelial cells, in different

tumour types? If not, how do the authors propose that the relative gene expression levels observed for specific cell types can be fairly compared between platforms? Would normalisation approaches such as normalisation of total transcript number or normalisation of housekeeping genes improve cross-platform comparability? Could this be done using the current dataset?

5. While the discussion provides a thorough summary of the results, the study would benefit from a more concise and clearly defined conclusion that explicitly highlights the key findings. Specifically, the authors should: Clearly state what new insights have been gained beyond previous knowledge. Provide practical recommendations on which spatial transcriptomics platform is best suited for different research applications. Identify key areas for future improvement in this field. Currently, these aspects are scattered throughout the discussion instead of being clearly formulated as a formal conclusion. A better structured and focussed conclusion would increase the impact of the manuscript and help readers to better understand its significance.

6. Captions should be more detailed to improve clarity and interpretability. In several cases, it is not entirely clear what specific aspects certain figure components represent. Given the importance of the figures in conveying the key findings, they should be self-explanatory and provide sufficient context without the reader having to constantly refer back to the main text.

Reviewer #2

(Remarks to the Author)

This manuscript presents a highly valuable study comparing spatial transcriptomics technologies using the same tissue and serial sections. Notably, the study evaluates the latest platforms—VISIUM HD, CosMx 6K, and Xenium 5K—which are expected to be widely used in the coming years. The findings provide crucial insights for researchers conducting spatial transcriptomic analyses and will likely be widely cited in the field.

However, while the manuscript effectively presents the results, it lacks a sufficient discussion on the differences between these technologies and the potential impact of these differences on the analysis. I strongly recommend expanding the discussion section to address these aspects in more detail.

Major Comments:

Impact of Gene Coverage and Specificity on Biological Pathway Analysis

VISIUM HD and Stereo-seq capture a large number of genes, whereas CosMx and Xenium analyze approximately 6,000 and 5,000 genes, respectively, with only about half of these genes overlapping.

The relatively lower specificity of CosMx may significantly influence downstream analyses, particularly biological pathway enrichment. Differences in gene coverage and specificity may affect the detectability of biological pathways, potentially altering the biological conclusions drawn from the same tumor regions.

This issue is particularly critical for pathways involving lowly expressed genes, such as IFN response pathways. Lower specificity could obscure the detection of these biologically important pathways, impacting the overall interpretation.

Providing supporting data and discussing these differences in detail would greatly enhance the manuscript.

Variations in Sensitivity, Specificity, and Concordance Across Cancer Types

The manuscript reports differences in sensitivity, specificity, and concordance across different cancer types. A more detailed discussion of these findings would provide valuable insights into their implications.

Minor Comments:

Figure 1b: VISIUM HD appears to be from a different tissue section compared to CosMx and Xenium. Is this correctly analyzed as a serial section? If possible, analyses should be conducted using tissue sections with more similar HE staining patterns to ensure comparability.

Figure 1g: When discussing sequencing-based sensitivity, sequencing saturation is an important parameter and should be presented.

Figure 2i: The manuscript states:

"In COAD, we identified TLS-like structures and analyzed the spatial distribution of associated transcripts. Concordance between ST and CODEX data for MS4A1 (B cell marker) and CD3E (T cell marker) suggests that ST platforms could recapitulate the cellular organization in TME (Fig. 2i)."

However, TLS structures do not appear to be clearly depicted in Stereo-seq or CosMx. If the claim is that all methods detect TLS similarly, objective numerical data should be provided to support this statement.

I appreciate the authors' efforts in conducting this comprehensive study and believe that addressing these points will further enhance the manuscript's impact.

Best regards,

Reviewer #3

(Remarks to the Author)

Ren et al present an impressive body of work entitled Systematic Benchmarking of High-Throughput Subcellular Spatial Transcriptomics Platforms. The authors have meticulously benchmarked spatial transcriptomics platforms at a time when

these technologies are starting to become more widely adopted, especially in translational research. Of note is that the authors conducted the benchmarking on three different tumor samples obtained from patients, and great care was taken to conduct sample preparation at a similar time frame. This greatly enhances the validity of the findings, since benchmarking on publicly available datasets provided by manufacturers are often from different samples which introduces a large degree of variability.

In order for the work to be publishable, we recommend the authors revise the manuscript according to the concerns raised below.

Major Concerns:

- 1) The title should indicate that the benchmarking was done on human cancer samples.
- 2) In the introduction, more detail about probe number in current technologies can be provided, and the advance to increase to 5k and 6k in new technologies.
- 3) Access to the raw data is not provided. It is becoming a standard that data be made available to view via online repositories (such as BioImageArchive) for accurate review. Although the SPATCH viewer the authors have created is impressive, the H&E tile scans are compressed to such an extent that histopathological interpretation is made impossible. Raw data download from SPATCH is also locked until publication.
- 4) Since the database the authors have created has such immense value to the field, there could be more discussion on the use of SPATCH, and how it could be leveraged to unlock current limitations in the field.
- 5) No information is provided in the supplementary on the patients from which the samples are pertained (age, sex, comorbidities etc.)
- 6) Figure 1: Visium FFPE and Xenium 5K are serial sections but the tissue looks very different. However, Xenium 5K and CosMx 6K are serial sections and do look serial. Is the scale different? What is causing this discrepancy? The CODEX data with anti-EpCAM antibody should also be included here.
- 7) Line 164: The authors don't elaborate on why specifically 8- μ m resolution was used for comparison when conducting comparisons for lineage markers.
- 8) Line 223: Further explanation should be given regarding the sensitivity benchmarking of sST technologies. Is the detection of non-human transcripts related to sequencing errors?
- 9) Assessing the number of negative probes in necrotic regions can be done to more accurately validate non-specific binding. More negative probes would be expected here.
- 10) For the evaluation of marker gene detection (Supp Figure 2d), the COAD data should be included in this comparison regardless of the irregular organization of cancer cells.
- 11) The variability in the Visium HD FF data is concerning, especially if there is a possibility of RNA degradation in the samples. There, it is strongly recommended that the Visium HD FF data is not included in the study.
- 12) Line 218-219: The conclusion that a high proportion of reads with barcodes not matching the mask file indicates a more significant loss of spatial information (Supplementary Fig. 4e) is not entirely accurate, since the mask file in this case was only the nuclear stain and the unmatched barcodes could be from the cytosolic component of cells.
- 13) Line 244-246: Extraction of high-quality calls and re-evaluating the negative control ratio was only conducted for CosMx 6K, the mitigation of bias for low-quality signals should also be conducted for Xenium 5K (Supplementary Fig. 5c).
- 14) The concordance variation observed between ST and CODEX could partially be described by the slight difference in cellular distribution of the adjacent section. There could also be artifacts introduced due to 'warping' of pixels during alignment. It would enhance the confidence of the results if the registered overlays of these sections are shown in the supplementary.
- 15) Line 285: The conclusion that iST platforms exhibited better concordance with CODEX than sST platforms (Supplementary Fig. 6c) does not have statistical merit. Visually there is more concordance in terms of structural representation, but this is not reflected in the correlation analysis.
- 16) Paragraph 288 – 295 seems out of context. Further detail should be provided as to the relevance of this finding?
- 17) The cell segmentation results are hard to interpret. Image quality for Fig 3a is very low, and given that cell boundary stains were used, this should be included in the image panel, along with the boundary segmentation to more clearly represent the differences.
- 18) Cell segmentation stains for Xenium are not specified in the methods.
- 19) It appears that the manual segmentation was only conducted on nuclei and not on entire cells? Representative Fig 3a does not make this clear. No mention or description is given in the methods on how manual segmentation was done or in which software. Was this done by thresholding or manually drawing outlines? In the methods, it is specified that the ST and CODEX correlations made use of DAPI for 'cell segmentation' using StarDist and QuPath. Was this the manual approach referred to for the comparison in Fig 3 as well? Can this truly be called 'ground truth'?
- 20) Line 313: iST platforms had a higher proportion of transcripts confined within the segmented cells compared to the sST platform (Supplementary Fig. 7f). Authors should clearly state whether this was comparing whole cell segmentation or nuclei segmentation.
- 21) Throughout the entire paper, the term cell segmentation is used, even though it is often nuclei segmentation. This should be made consistent and clear so as to prevent confusion.
- 22) Although official segmentation tools are not provided by the supplier for Visium HD FFPE, the existence of many open source segmentation tools still make it possible to do a fair comparison. Since H&E images are available, simple cell segmentation in QuPath could be applied to give an indication of how the transcript counts compare to other technologies.
- 23) Line 346: Evaluation of Cell Clustering. In addition to showing the clustering visually (Figure 4d), a quantitative metric should be used to compare the accuracy. Perhaps a Jacard index or similar. The visual only comparison is hard to interpret.
- 24) Line 778: Diffusion distances – Diffusion must also be occurring within the tissue. In the discussion, it would be good to see a comparison of FFPE and Frozen in general with regards to diffusion distance.
- 25) Optical crowding is not mentioned, but must be impacting the large 5k and 6k panels. This should be included in the discussion or introduction.
- 26) Line 506: We strongly suggest removing the statement about 'critical contribution to the field' - this should be implied in

quality and depth of the study.

Editing Concerns :

- 1) Remove website from abstract, this should be in the main text.
- 2) Graphs are small to illegible.
- 3) No scale bars are provided on any of the images.
- 4) Not all abbreviations are defined (for example SPATCH and UMI)
- 5) Figure 1: use same colors across all graphs (ie Fig 1 e and f)
- 6) Supplementary Figure 1: At least one of the chosen ROIs should be enhanced to so it can be validated whether they are indeed representative of cancer cells with similar morphology.
- 7) Figure 2c is not displaying a ratio (as mentioned in text) but rather displays counts.
- 8) Fig 2e: Figure legend on y-axis does not show distance metric (was it in microns or pixels?)
- 9) Figure 2g: the blue stain from the CODEX is barely visible. A using a more pronounce color would be suggested.
- 10) Figure 4a, a legend for the clusters is missing.

Reviewer #4

(Remarks to the Author)

Version 1:

Reviewer comments:

Reviewer #1

(Remarks to the Author)

I would like to thank the authors for their thoughtful and thorough responses to my comments. I appreciate the detailed justifications provided, the inclusion of additional analyses, and the clarifications throughout the revised manuscript. The authors have successfully addressed all my major concerns, and the quality and clarity of the work have improved substantially. I have no further comments regarding the manuscript.

Reviewer #2

(Remarks to the Author)

Thank you for your detailed reply. Regarding Response Figure 2, I feel it does not address the core of my concern. In spatial transcriptomics, the biological pathways that play important roles in the tissue of interest are often a primary focus. Therefore, differences in gene coverage and the lower specificity of certain panels can have a substantial impact on biological pathway analysis. In my view, analyzing different locations on separate tissue slices or restricting the analysis to only overlapping genes is unnecessary. Instead, what matters is how much the list of enriched biological pathways differs when GSEA is performed on the same position across serial sections. Because spatial transcriptome data are sparse, I expect that CosMx's lower specificity will have a large effect on pathway enrichment outcomes.

To perform this analysis, high-quality serial sections are needed. From the figures you presented, it appears that the Xenium and CosMx samples are from nearly identical positions, suggesting it should be possible to conduct GSEA at the same location for these two platforms. For other platforms, please include comparable analyses if same-position data are available. I am not looking for similar results across platforms; rather, understanding how different the enriched pathway lists are—and which pathways each method can detect more easily—would provide valuable insights to users. As such, I do not believe Response Figure 2 is essential. I would appreciate it if you could consider creating the figure I have requested, even as a supplemental figure. Regarding the other parts of your response, you have provided sufficient answers to my questions.

Reviewer #3

(Remarks to the Author)

We thank the authors for applying our recommendations and amending their manuscript accordingly. In general the majority of our concerns have been addressed and additional analyses have been completed and supplementary data provided. However, we need to highlight the following inconsistencies pertaining mostly to the image data and segmentation data. We therefore recommend these minor revisions prior to publication.

Response 6:

The newly generated images in Supplementary Figure 2a for Xenium and CosMx are extremely dim and tissue outlines can't be viewed accurately. Please enhance the contrast so the tissue is actually visible.

Response 8:

Given that Stereo-seq captures non-human transcripts so sensitively, we recommend the authors identify whether these are associated with microbial genomes to showcase that any possible contamination will be reflected in the Stereo-seq data. This will allow for a precautionary recommendation to readers/potential users that special care must be taken when conducting slide preparation for Stereo-seq specifically.

Response 14:

Again the representative tissue images for the CosMx and Xenium data are dim to illegible. No scale bars are present for Supplementary Image 6 either in this context. Please ensure that these are amended prior to publication.

Responses 17:

While the authors do provide updated images showing cell membranes, the merged images in Response figure 3.7a do not show an actual merged between nuclei and cell membrane segmentations. Furthermore, the term 'annotation' is being used here when it should be 'segmentation' to avoid confusion. A more zoomed in representation of outer membrane vs nuclei segmentation would be recommended for a journal of this calibre (as the authors have done in Supplementary Figure 13).

Responses 19-20:

We have concern regarding the inconsistency of segmentation data. In Figure 3 nuclei segmentation based efficiency is compared between technologies based on manual nuclei segmentation which makes sense due to differences between technologies. However, in the following datasets the authors then assign transcripts to whole cell segmentations based either on nuclei dilation method (expansion from StarDIST) for Stereo-seq data and then Xenium and CosMx cell boundary based segmentation. Doing comparison on segmentation efficiency on nuclei only is made void if the rest of the downstream analysis is actually done on whole cells.

Since a dilation approach is used to capture transcripts in whole cells for Stereo-seq data, the description and axis legend of Figure 10c must change from 'transcripts detected per cell/nucleus' to only 'transcripts detected per cell'. It is biologically inaccurate to state that both cell and nuclei encapsulate the same compartment of tissue.

Software version for Labelme must also be provided.

Reviewer #4

(Remarks to the Author)

Version 2:

Reviewer comments:

Reviewer #2

(Remarks to the Author)

Thank you for clearly understanding the intent of my previous comments and for performing the additional analyses. The revised comparison using anatomically matched regions across Visium HD FFPE, CosMx 6K, and Xenium 5K fully addresses my core concern and provides valuable insight into platform-specific differences in pathway detection. I appreciate your thoughtful work, and I am satisfied that my questions have been thoroughly addressed.

Reviewer #3

(Remarks to the Author)

Thank you for addressing my comments.

Reviewer #4

(Remarks to the Author)

Response letter to Review comments of the manuscript “Systematic Benchmarking of High-Throughput Subcellular Spatial Transcriptomics Platforms” (NCOMMS-25-06494-T).

RESPONSES TO REVIEWERS

Reviewer #1:

Summary of the findings

The study aims to compare high-throughput spatial transcriptomics (ST) platforms, comparing sequencing-based (sST) and imaging-based (iST) approaches. While the fundamental trade-offs between these methodologies — iST platforms, which offer superior spatial accuracy but limited transcriptome-wide profiling, and sST platforms, which offer broader coverage at the expense of spatial precision — are well recognized, this study helps to refine existing knowledge through more rigorous comparisons. In particular, it minimizes batch effects through standardized conditions, includes multi-omics validation by single-cell RNA sequencing (scRNA-seq) and CODEX proteomics, and provides a publicly available dataset (SPATCH). While these efforts improve methodological rigor, the study does not fundamentally redefine our understanding of the performance of ST platforms. The main conclusions regarding the trade-offs between platforms align closely with previous studies, which raises the question of whether the primary value of the study lies in new insights or simply a higher quality benchmark dataset. The following points highlight areas where additional justifications, methodological refinements and clearer conclusions could improve the manuscript.

Response: We thank the Reviewer for this concise summary of our work. As shown below in our responses, we have carefully addressed the concerns raised and find that the constructive comments have helped to improve the manuscript in this revised version.

Major concerns and limitations

1. The different spatial transcriptomics platforms in this study used either fresh-frozen (FF) or formalin-fixed paraffin-embedded (FFPE) tissue sections. Given that all platforms support FF, can the authors justify why different fixation methods were chosen for the different platforms? This choice introduces additional variability as FF and FFPE samples undergo different preservation processes that may affect RNA integrity and gene expression profiles. In addition, the need to cut different tissue blocks introduces further variability and may affect the efficiency of direct cross-platform comparisons. Given these factors, how do you account for and mitigate potential biases introduced by these methodological differences? If not, how might this affect direct platform comparisons?

Response: We sincerely thank the Reviewer for the thoughtful comment regarding the selection of embedding methodologies. We fully agree that using consistent

embedding strategies would further optimize cross-platform comparisons. However, we selected different fixation methods based on practical and technical considerations at the time of study design (April 2024), as outlined below: (1) Although Visium HD is now available for both FFPE and FF embeddings, the FF version had not yet been officially released when we initiated our experiments (April 2024). At that time, Stereo-seq exclusively supported FF embedding and was not compatible with FFPE. (2) FFPE is the predominant embedding method used in clinical biobanks due to its superior long-term preservation and stability. Benchmarking on FFPE samples therefore increases clinical relevance and broadens the applicability of our findings. (3) Moreover, FFPE embedding better preserves tissue morphology, pathological features, and nuclear structures (**Response Figure 1.1a-d**), which are especially critical for accurate cell segmentation and downstream spatial analyses in high-resolution platforms. For these reasons, we selected FFPE embedding for Visium HD, CosMx 6K, and Xenium 5K, aiming to provide valuable guidance to the large community of users working with FFPE-preserved clinical samples.

In terms of sample preparation, to minimize RNA degradation, tissue samples were immediately sectioned following collection, with FFPE and FF tissues processed in parallel. FF blocks were stored at -80°C and transported on dry ice, while FFPE samples were preserved under low-temperature, low-humidity conditions to maintain RNA integrity. The maximum interval between tissue collection and spatial transcriptomics experiments was limited to three months to reduce the risk of RNA degradation. Prior to initiating spatial transcriptomics experiments, all samples underwent quality control assessments recommended by the respective platform manufacturers. These evaluations confirmed that RNA integrity met the required thresholds and showed no evidence of significant degradation.

In terms of study design, each tumor sample was carefully divided into three equal sections: one section was used for scRNA-seq, while the remaining two were embedded using FF and FFPE protocols, respectively. This design ensured a high level of consistency between the scRNA-seq reference and the tissue blocks processed for spatial transcriptomics. We agree that cross-block comparisons can introduce variability. To minimize this, we also evaluated tumor regions with similar tissue morphology for cross-platform comparisons. Additionally, to avoid over-reliance on direct comparisons across spatial transcriptomics platforms, we implemented a rigorous framework that integrated orthogonal omics modalities, such as scRNA-seq and CODEX, derived from adjacent sections or tissue blocks as reference standards. CODEX data derived from adjacent sections was used as a spatial reference for each platform, enabling robust benchmarking of transcript localization, cell type annotation, and spatial clustering. To further ensure consistency, all tissue sectioning for FFPE samples was performed by a single technician, maintaining uniform slice quality across platforms.

Collectively, our sample preparation, experimental design, and analytical strategies

were carefully optimized to minimize variability arising from different embedding methods. The use of FFPE embedding particularly enhances the clinical relevance of our study, given its widespread adoption in biobank repositories. We hope this detailed explanation adequately addresses the Reviewer's concerns and clarifies the rationale behind our methodological choices.

Response Figure 1.1. Comparison of tissue morphology and nuclear staining between FFPE and FF samples.

a-b. H&E staining of colorectal cancer samples processed with Visium HD FFPE (**a**) or Visium HD FF (**b**) protocols. Magnified views of the boxed regions are shown on the right.

c-d. DAPI staining of adjacent sections corresponding to panels **a** and **b**, respectively. Magnified views of the boxed regions are shown on the right.

2. The study finds that different spatial transcriptomics platforms perform better in certain tumour types. What factors do the authors believe have contributed to these platform- and sample-dependent variations? Also, do the authors believe that certain platforms are inherently better suited to certain tumour microenvironments, or do these differences arise primarily from experimental conditions and sample processing?

Response: We appreciate the Reviewer's insightful comments regarding the platform- and sample-dependent variations observed in our study. Below, we outline the key factors that may contribute to these variations:

(1) Variation in gene and transcript detection across platforms

We observed platform- and tissue-specific differences in the number of genes and transcripts detected per bin, particularly for CosMx 6K and Xenium 5K. For example, Xenium 5K showed superior performance in HCC, whereas CosMx 6K outperformed in OV when considering all detected genes (**Response Figure 1.2a**). These variations are likely driven by the differences in the gene panels employed by each platform, as

well as sample-specific variability. Supporting this, when restricting the analysis to shared genes, Xenium 5K consistently detected more genes and transcripts per bin across both the 10 selected tumor regions and shared areas within FFPE serial sections (**Response Figure 1.2b**).

Response Figure 1.2. Comparison of bin-level transcript and gene detection sensitivity across spatial transcriptomics platforms.

a-b. Log₂-transformed transcript and gene counts per bin across two types of regions: ten morphology-matched regions within each platform (left) and overlapping regions shared among FFPE-based platforms (right), using all detected genes (**a**) and restricted to CosMx 6K and Xenium 5K using shared genes (**b**).

(2) Transcript and gene counts in manually segmented cell nuclei

We also compared gene and transcript counts per manually segmented cell nucleus for CosMx 6K and Xenium 5K. Interestingly, CosMx 6K showed better performance in COAD, while Xenium 5K outperformed in HCC and OV when analyzing the same set of common genes (**Response Figure 1.3a**). These findings were inconsistent with bin-based results (**Response Figure 1.2b**). Upon further investigation, we identified substantial differences in nuclear sizes across certain matched regions, which likely confounded the analysis and impacted the observed gene and transcript counts (**Response Figure 1.3b-d**). Given this limitation, we have removed this analysis from the revised manuscript.

Response Figure 1.3. Comparison of nucleus-level metrics between CosMx 6K and Xenium 5K.

a. Box plots showing log₂-transformed total transcript (top) and gene (bottom) counts per manually segmented nucleus across five manually annotated regions of interest (ROI).

b-d. Per-ROI comparisons of nuclear size (**b**), gene count (**c**), and transcript count (**d**) for ROI1 to ROI5 across each cancer type.

(3) Concordance between transcript localization and CODEX-derived spatial distributions

We also observed platform- and tissue-specific variability in the spatial alignment between ST-derived transcript localization and raw CODEX staining signal intensity. One major contributing factor was background noise in the CODEX data, which varied across tissue types (**Response Figure 1.4a-b**). To address this, we refined our analysis by correlating transcriptomic data with CODEX-derived cell type counts, rather than with continuous intensity signals. We also standardized the cell-type signatures used across tissues by selecting the top 15 differentially expressed genes for each cell type from scRNA-seq data (COAD, HCC, and OV) and retaining genes shared across at least two tissue types. By intersecting these gene sets with common genes shared across spatial transcriptomics platforms, we controlled for platform differences in gene coverage. This updated approach improved alignment across platforms. Overall, we found that Visium HD FFPE outperformed Stereo-seq v1.3, and Xenium 5K outperformed CosMx 6K in terms of spatial concordance with CODEX

(**Response Figure 1.4 c-f**). These revisions have been incorporated into the 'Results' and 'Methods' sections of the manuscript, which now read as follows:

Results section:

"Visium HD FFPE and Xenium 5K showed higher concordance with CODEX than Stereo-seq v1.3 and CosMx 6K, respectively (**Supplementary Fig. 7e**). For epithelial cells, which are highly abundant in both COAD and OV, inter-platform differences were relatively minor (**Supplementary Fig. 7f**).

Global concordance analysis indicated higher consistency for Visium HD FFPE and Xenium 5K over Stereo-seq v1.3 and CosMx 6K, respectively (**Fig. 2h**). As observed previously, the differences for epithelial cells among platforms remained minimal (**Fig. 2i**)".

Methods section:

"We evaluated the concordance between CODEX-based cell annotations and ST-derived features including: (1) individual marker gene expression, (2) cell-type-specific gene signatures, and (3) cell type abundance. To construct gene signatures, we selected the top 15 differentially expressed genes for each cell type from matched scRNA-seq data, and retained those present in at least two tissue types. To mitigate platform-specific gene panel differences, we intersected these genes with common genes shared by all four ST platforms. The shared regions of ST and CODEX data were binned at multiple spatial resolutions (100, 200, 300, 400, and 500 μm). For each spatial bin, we quantified ST-derived features at all three levels, alongside corresponding cell counts inferred from CODEX. Pearson correlation coefficients were then computed across all spatial bins to assess the spatial concordance between the ST-derived features and CODEX annotations".

Response Figure 1.4. Correlation between spatial transcriptomics and CODEX-derived cell-type distributions.

a–b. Spatial distribution of CODEX staining intensities for CD4 (**a**) and CD8 (**b**) in representative tissue sections.

c–d. Spatial correlation between CODEX-inferred cell counts and ST-derived marker gene expression for different cell types over the spatial grids. Panel **c** shows the correlations for immune and stromal marker genes, while panel **d** shows the correlations for epithelial marker gene. Pearson correlation coefficients are reported. Error bars represent standard error of the mean (SEM) for correlations obtained under different grid sizes.

e–f. Spatial correlation between CODEX-inferred cell counts and ST-derived signature scores for different cell types over the spatial grids. Panel **e** shows the correlations for immune and stromal signature scores, while panel **f** shows the correlations for epithelial signature scores. Pearson correlation coefficients are reported. Error bars represent SEM for correlations obtained under different grid sizes.

(4) Expression correlation between scRNA-seq and spatial transcriptomics data across cell types

We observed notably lower correlations between scRNA-seq data and spatial transcriptomics profiles in HCC samples for Stereo-seq v1.3 and Visium HD FFPE (**Response Figure 1.5a**). This discrepancy is likely due to the lower accuracy of cell segmentation and cell-type resolution in sequencing-based platforms, which do not leverage high-resolution imaging for boundary delineation. For example, multiple cell types in sequencing-based platforms displayed elevated expression of the hepatocyte marker gene *ALB*, leading to less distinct cell-type profiles and, consequently, lower concordance with scRNA-seq references (**Response Figure 1.5b**).

Response Figure 1.5. Concordance between spatial transcriptomics and scRNA-seq for cell-type-specific expression profiles.

a. Pearson correlation coefficients between ST-derived and matched scRNA-seq-derived expression profiles across annotated cell types in COAD, HCC, and OV. Each box represents the distribution of correlation values per platform.

b. Dot plots showing expression levels of selected cell-type marker genes in Stereo-seq v1.3 and Visium HD FFPE datasets. Dot size indicates the proportion of cells expressing each gene within the annotated cell type. Color intensity reflects average expression.

(5) Discrepancies in spatial clustering performance in HCC samples

In spatial clustering evaluations, CosMx 6K exhibited substantially higher concordance with CODEX in HCC samples compared to other platforms. However, this observation was primarily driven by technical artifacts in the CODEX staining of the HCC sample. In particular, we observed high staining variability across different tissue sections, especially in the FOXP3 and CD4 channels. Elevated background noise was detected in the CODEX slide adjacent to the CosMx 6K section. In contrast, no notable differences were observed in these channels between CosMx 6K and Xenium 5K adjacent CODEX sections in COAD and OV samples (**Response Figure 1.6a**). To mitigate the impact of these artifacts, the affected channels were excluded from the updated HCC clustering analysis (**Response Figure 1.6b**). Additionally, we found that spatial clusters derived from transcriptomics and CODEX data did not demonstrate a strict one-to-one correspondence. To address this discrepancy, we assigned each CODEX cluster its most correlated transcriptomic cluster and computed the average

correlation coefficients across all matched cluster pairs. This approach enabled a more robust cross-modality alignment (**Response Figure 1.6c**). These updates have been incorporated into the revised 'Results' and 'Methods' sections, which now read as follows:

Results section:

“Overall, comparable clustering concordance with CODEX was observed across ST platforms, with the exception of Stereo-seq v1.3 in HCC, which exhibited low concordance (**Fig. 5b**)”.

Methods section:

“For HCC, CD34, CD4, FOXP3, and HLA-A channels were excluded from CODEX clustering due to quality concerns. To investigate the clustering concordance between CODEX and ST data, we computed cell proportions within spatial grids (100, 200, 300, 400, and 500 μm) for each cluster and assessed their correlations across all grids. For each CODEX cluster, we identified the best-matching ST cluster based on the maximal correlation, and the final metric was defined as the average correlation across all matched cluster pairs”.

Response Figure 1.6. Spatial clustering concordance between spatial transcriptomics and CODEX.

a. Spatial distribution of CODEX signals for CD4, FOXP3, CD34, and HLA-A in adjacent sections corresponding to CosMx 6K and Xenium 5K datasets. Color scale reflects staining intensity.

b. Comparison of spatial clustering results from ST (top) and CODEX (bottom) in HCC samples across four platforms.

c. Quantification of clustering concordance between ST and CODEX datasets using Pearson correlation. Error bars represent SEM across different grid sizes.

3. The Xenium In Situ Cell Segmentation Kit (10x Genomics) and NanoString's CosMx™ Human Universal Cell Segmentation Kit provide advanced multimodal segmentation by staining intracellular RNA, proteins, cell borders and nuclei for improved cell segmentation. Given that segmentation accuracy critically affects

downstream spatial transcriptome analyses, can the authors explain why they did not include these advanced segmentation kits in their assessment? Do they anticipate that the use of such multimodal approaches would increase the accuracy of transcript localization within cells and improve the overall representation of cell shape, particularly for imaging-based ST platforms? Furthermore, have alternative segmentation refinements been attempted post hoc (e.g., deep learning-based segmentation) to mitigate challenges in complex cell types? Additionally, how do you think these methods would perform in multinucleated cells (e.g., skeletal muscle fibers, syncytial trophoblasts, osteoclasts), where standard nuclear-based segmentation may be insufficient to capture their true morphology and transcript distribution?

Response: We thank the Reviewer for the insightful comments regarding the use of advanced multimodal cell segmentation kits and their potential impact on segmentation accuracy. Both CosMx 6K and Xenium 5K experiments in our study employed these advanced segmentation tools. Below, we discuss the effects of multimodal segmentation on cell morphology representation, transcript localization, and the handling of complex cell types.

(1) Cell shape

The Xenium 5K platform provides paired nuclear and whole-cell segmentation, enabling a more complete representation of cell morphology. In HCC samples, we observed that hepatocytes exhibited markedly irregular cell shapes relative to their nuclei (**Response Figure 1.7a**). Moreover, when comparing across cell types, hepatocytes showed greater morphological complexity than T cells (**Response Figure 1.7a**). Quantitative analysis using morphological metrics, including solidity (where higher values indicate more convex shapes) and circularity (where higher values indicate more rounded shapes), confirmed that hepatocytes had lower scores under whole-cell segmentation, reflecting their irregular geometry (**Response Figure 1.7b**). These results highlight the importance of multimodal segmentation for accurately capturing cell shape.

(2) Transcript localization

Multimodal whole-cell segmentation also improves transcript localization (**Response Figure 1.7c**). In both Xenium 5K and CosMx 6K datasets, whole-cell segmentation retained significantly more transcripts from key identity marker genes, compared to nuclear segmentation generated using StarDist¹ (**Response Figure 1.7d**). This underscores the advantage of whole-cell boundaries in capturing transcripts that would otherwise fall outside nuclear masks.

(3) Use of alternative segmentation refinements

We did not apply additional post hoc refinement methods to the segmentation results. Both the Xenium 5K and CosMx 6K platforms implement state-of-the-art segmentation algorithms. Xenium platform uses custom deep learning models trained on multi-channel staining images from diverse tissue types and preservation methods (FF and

FFPE). CosMx platform utilizes the enhanced CellPose² algorithm, which integrates signals from membrane proteins, nuclei, and RNA, and has been trained on over 50 million single cells across hundreds of tissue types. Given the robustness and extensive validation of these built-in methods, we consider them sufficient for accurate cell delineation. Additionally, we intentionally avoided manual or alternative refinements to maintain consistency with the platform-generated outputs, which represent the default and most accessible form of the data for typical users. Reprocessing the segmentation results would introduce additional complexity and reduce comparability with standard platform usage.

(4) Multinucleated cell segmentation

Whole-cell segmentation is particularly advantageous for identifying multinucleated cells, such as neutrophils, skeletal muscle fibers, syncytial trophoblasts, and osteoclasts, that are not well captured by nuclear segmentation alone. In the Xenium 5K HCC data, we identified a substantial number of multinucleated cells (**Response Figure 1.7e-f**). Our analysis showed that neutrophils and hepatocytes exhibited higher frequencies of multinucleation, consistent with biological expectations (**Response Figure 1.7g-i**).

These changes are reflected in the 'Results' section under 'Evaluation of cell clustering and cell type annotation'.

Response Figure 1.7. Whole-cell segmentation improves cell shape delineation and transcript assignment accuracy.

a. Representative field of view from the Xenium 5K HCC sample showing fluorescence image with DAPI and membrane staining (left) and corresponding segmentation results (right), with T cells highlighted in green and hepatocytes in pink.

b. Comparison of mask shape metrics between whole-cell and nuclear segmentation for hepatocytes and T cells in the Xenium 5K HCC sample.

c. Spatial localization of CD3E, CD4, and CD68 transcripts within membrane-based (whole-cell) versus nucleus-based boundaries.

d. Bar plots showing the proportion of transcripts assigned to either nuclear or whole-cell segmentation masks across cancer types in CosMx 6K and Xenium 5K datasets. Nuclear masks were generated using StarDist. The cluster of differentiation genes were used in this analysis.

e. Representative field of view from the Xenium 5K HCC sample illustrating cells with

multiple nuclei. Membrane and nuclear boundaries are outlined.

f. Histogram showing the distribution of cells by number of nuclei per cell in the Xenium 5K HCC dataset.

g. Proportion of multinucleated cells across annotated cell types in the Xenium 5K HCC sample.

h-i. Representative examples of multinucleated neutrophils (**h**) and hepatocytes (**i**) across three regions of interest (ROIs), showing membrane and nuclear boundaries along with localization of CSF3R (neutrophil marker) and SAA4 (hepatocyte marker) transcripts.

4. The study evaluates marker gene detection and molecular capture efficiency on different platforms, but it is unclear whether relative gene expression within the same cell types on different platforms was explicitly compared. Given that different ST platforms have different efficiencies in capturing transcripts, did the authors attempt to normalise gene expression across platforms to allow direct comparisons between specific cell types, such as endothelial cells, in different tumour types? If not, how do the authors propose that the relative gene expression levels observed for specific cell types can be fairly compared between platforms? Would normalisation approaches such as normalisation of total transcript number or normalisation of housekeeping genes improve cross-platform comparability? Could this be done using the current dataset?

Response: We thank the Reviewer for highlighting this important point. As suggested by the Reviewer, to enable fair comparisons of relative gene expression levels across platforms, we applied two normalization strategies: (1) normalization based on the total counts of genes shared across all five platforms (**Response Figure 1.8a**), and (2) normalization using the mean expression of a curated set of housekeeping genes (**Response Figure 1.8b**). Following normalization, we compared the expression levels of canonical marker genes across platforms within specific cell types, including CD34 (endothelial cells), COL5A2 (fibroblasts), CD68 (macrophages), and CD8A (T cells). Among all platforms, Xenium 5K was the only one that consistently achieved marker gene expression levels comparable to those observed in the matched scRNA-seq data, indicating more accurate relative quantification.

These updates are now reflected in the 'Results' section, which read as: "Xenium 5K most closely recapitulated scRNA-seq profiles for canonical cell type markers, including CD34 (endothelial cells), COL5A2 (fibroblasts), CD68 (macrophages), and CD8A (T cells) (**Supplementary Fig. 11e, f**)".

Response Figure 1.8. Normalized expression of cell-type-specific marker genes across platforms.

a-b. Expression levels of selected canonical marker genes, normalized by either the total counts of genes shared across all platforms (**a**) or the mean expression of a curated housekeeping gene set (**b**). Each box shows the expression of the corresponding marker gene within its annotated cell type: CD8A (T cells), CD68 (macrophages), CD34 (endothelial cells), and COL5A2 (fibroblasts).

5. While the discussion provides a thorough summary of the results, the study would benefit from a more concise and clearly defined conclusion that explicitly highlights the key findings. Specifically, the authors should: Clearly state what new insights have been gained beyond previous knowledge. Provide practical recommendations on which spatial transcriptomics platform is best suited for different research applications. Identify key areas for future improvement in this field. Currently, these aspects are scattered throughout the discussion instead of being clearly formulated as a formal conclusion. A better structured and focused conclusion would increase the impact of the manuscript and help readers to better understand its significance.

Response: We thank the Reviewer for the valuable suggestion. We have revised the 'Discussion' section to provide a clearer and more concise conclusion. The revised section now explicitly summarizes the key findings, offers practical guidance for platform selection based on specific research needs, and includes a dedicated outlook on future directions. The added contents now read as:

- (1) New insights have been gained beyond previous knowledge
- “At the technical level, both sST and iST platforms demonstrated distinct strengths and limitations. Among the sST platforms, Visium HD FFPE outperformed Stereo-seq v1.3

in transcript detection sensitivity and specificity, likely due to its targeted transcript capture strategy and reduced transcript diffusion. Within the iST platforms, Xenium 5K exhibited stronger gene-wise correlations with scRNA-seq and more effective background signal control than CosMx 6K, which showed elevated signals for low-abundance genes. Notably, both Xenium 5K and Visium HD FFPE exhibited higher spatial concordance with CODEX-based protein maps, underscoring their spatial accuracy. Beyond transcript-level assessments, we also benchmarked cell segmentation performance. Xenium 5K achieved superior segmentation accuracy, minimizing transcript spillover between adjacent cells and enabling better resolution of single-cell boundaries. Importantly, it also captured more irregular cell morphologies, reflecting the well-established morphological heterogeneity among different cell types.

In downstream analyses, Xenium 5K demonstrated the highest accuracy in cell type annotation across all platforms, as evidenced by several key metrics. It achieved the greatest annotation consistency across multiple computational tools, exhibited cleaner marker gene expression, and showed the strongest concordance with adjacent CODEX protein data. Furthermore, Xenium 5K effectively delineated fine-grained spatial organization of specific cell types, such as immune cells within TLS-like regions and endothelial cells lining vascular structures. The use of multi-channel imaging enhanced annotation accuracy by enabling more precise cell boundary delineation and identification of multinucleated cells, a feature often missed by nucleus-only segmentation strategies. This multimodal segmentation approach improved transcript assignment to the correct cells, thereby providing more accurate cellular transcriptomic profiles. In contrast, Visium HD FFPE, while demonstrating high transcription detection sensitivity and specificity, showed reduced cell type annotation due to the absence of reliable single-cell segmentation. For spatial clustering, all platforms successfully recapitulated large-scale tissue architecture. However, only Visium HD FFPE and Xenium 5K accurately delineated continuous tumor boundaries. In regional pathway enrichment analyses, both gene panel size and expression specificity proved critical, as artificial reduction of panel size or introduction of expression noise markedly attenuated pathway enrichment signals”.

(2) Practical recommendations on which spatial transcriptomics platform is best suited for different research applications

“Our findings provide practical guidance for selecting ST platforms based on study objectives. For single-cell level analyses, iST platforms are preferable due to their use of multichannel staining and full-cell segmentation, which enable more accurate cell boundary delineation and reduce transcript spillover. Their targeted capture strategies also enhance the sensitivity and specificity of marker gene detection, supporting cell state inference. Among the iST platforms, CosMx 6K was more susceptible to background noise, which may lead to false detection of low-abundance genes and reduce annotation fidelity. In contrast, for spatial analyses focused on tissue-region level patterns, sST platforms may be more suitable due to their broader gene coverage, which improves sensitivity for pathway-level enrichment analyses. Within this category,

Visium HD FFPE offered better performance, driven by its reduced transcript diffusion and high detection accuracy. Finally, for applications involving host–microbe interactions, Stereo-seq v1.3 holds unique advantages due to its unbiased poly(A)-based capture, which allows for the detection of both human and non-human transcripts”.

(3) Key areas for future improvement in this field

“Looking ahead, each platform class faces distinct technical challenges. For iST technologies, expanding toward whole-transcriptome coverage while maintaining high specificity and detection efficiency is a critical frontier. However, optical crowding remains a limiting factor, as larger gene panels may reduce detection efficiency due to signal overlap⁴⁴. Current strategies, such as reducing the number of probes per gene and increasing imaging cycles, show promise but require further innovations in probe design, codebook optimization, and the elimination of background fluorescence suppression to maintain specificity. For sST platforms, improvements in spatial resolution, control of transcript diffusion, and accurate segmentation are needed to limit transcript leakage from neighboring cells. These may be addressed through advanced chip engineering and integration of DNA-staining-based segmentation methods. Importantly, permeabilization protocols remain a key determinant of transcript diffusion. Optimizing reagent formulations and incubation times can significantly reduce leakage²⁸. This consideration is especially critical for FFPE samples relative to FF samples, as FFPE embedding can compromise RNA integrity. The resulting RNA fragmentation increases the likelihood of transcript leakage during the permeabilization process. Additionally, protocols could be improved during fresh-frozen tissue preparation, as slow freezing of tissues can induce ice crystal formation and membrane disruption, exacerbating transcript diffusion artifacts. Beyond these platform-specific considerations, broader technological advances will shape the future of ST development as well. The development of three-dimensional spatial transcriptomics for thick tissue sections, relaxation of sample input constraints, progress in non-destructive spatial profiling, and integration with multiomic modalities will collectively expand the scope, accessibility, and impact of ST technologies in both research and clinical settings”.

6. Captions should be more detailed to improve clarity and interpretability. In several cases, it is not entirely clear what specific aspects certain figure components represent. Given the importance of the figures in conveying the key findings, they should be self-explanatory and provide sufficient context without the reader having to constantly refer back to the main text.

Response: We thank the Reviewer for this helpful suggestion. We have carefully revised the figure captions to enhance clarity and ensure that each figure is self-explanatory and interpretable without reference to the main text. For example, we explicitly specified (1) the definition of each data point in the box plots, and (2) whether the evaluation was performed at nucleus, cell, or bin level, as well as the segmentation

strategy applied for each ST platform in the revised figure captions.

Reviewer #2:

Summary of the findings

This manuscript presents a highly valuable study comparing spatial transcriptomics technologies using the same tissue and serial sections. Notably, the study evaluates the latest platforms—VISIUM HD, CosMx 6K, and Xenium 5K—which are expected to be widely used in the coming years. The findings provide crucial insights for researchers conducting spatial transcriptomic analyses and will likely be widely cited in the field. However, while the manuscript effectively presents the results, it lacks a sufficient discussion on the differences between these technologies and the potential impact of these differences on the analysis. I strongly recommend expanding the discussion section to address these aspects in more detail.

Response: We thank the Reviewer for providing a concise summary of our work. As detailed in the responses below, we have carefully addressed each of the concerns raised. The Reviewer's comments have significantly improved the manuscript in this revised version.

Major concerns and limitations

1. Impact of Gene Coverage and Specificity on Biological Pathway Analysis

VISIUM HD and Stereo-seq capture a large number of genes, whereas CosMx and Xenium analyze approximately 6,000 and 5,000 genes, respectively, with only about half of these genes overlapping. The relatively lower specificity of CosMx may significantly influence downstream analyses, particularly biological pathway enrichment. Differences in gene coverage and specificity may affect the detectability of biological pathways, potentially altering the biological conclusions drawn from the same tumor regions. This issue is particularly critical for pathways involving lowly expressed genes, such as IFN response pathways. Lower specificity could obscure the detection of these biologically important pathways, impacting the overall interpretation. Providing supporting data and discussing these differences in detail would greatly enhance the manuscript.

Response: We thank the Reviewer for highlighting the critical role of gene panel coverage and specificity in the interpretation of biological pathways. To address this concern, we evaluated the impact of gene coverage and expression specificity on pathway enrichment, using the interferon gamma signaling pathway as a representative example.

We first selected three $500 \times 500 \mu\text{m}$ regions enriched for T cells and three regions with sparse T cell infiltration from each of the four spatial transcriptomics platforms, based on adjacent CODEX data (**Response Figure 2.1a**). Using the R package clusterProfiler⁵, we performed gene set enrichment analysis (GSEA) to assess the interferon gamma pathway enrichment, an important pathway for immune biology and

is also clinically relevant, in the T cell-enriched regions compared to the T cell-depleted regions. All platforms showed pathway enrichment with varying degrees of significance, likely influenced by both gene panel difference and region selection (**Response Figure 2.1b**). Among all platforms, Visium HD FFPE demonstrated the most robust enrichment.

Given Visium HD FFPE's broader gene panel and previously demonstrated higher expression specificity, we used Visium HD FFPE data to systematically evaluate how gene panel size and detection specificity affect pathway enrichment. To assess the impact of gene coverage, we restricted the analysis to genes overlapping with the panels of CosMx and Xenium platforms, and repeated the GSEA analysis (**Response Figure 2.1c**). Notably, smaller panels often failed to detect enrichment, while larger panels consistently yielded significant results (**Response Figure 2.1d**).

We then extended this analysis using a simulation-based approach. From the Visium HD FFPE panel, we retained subsets of interferon gamma pathway genes of varying sizes and performed 100 rounds of downsampling followed by GSEA analysis. Enrichment scores increased with gene set size, confirming that greater gene coverage enhances pathway detectability (**Response Figure 2.1e**). As a negative control, we repeated the same procedure using housekeeping genes, which did not yield significant enrichment regardless of subset size (**Response Figure 2.1e**), supporting the specificity of the observed effect. To evaluate the impact of expression specificity, we added increasing levels of Gaussian noise to the gene expression values. As noise increased, enrichment scores declined and, in some cases, reversed direction (**Response Figure 2.1f**), highlighting the importance of accurate quantification for reliable pathway detection. Together, these results demonstrate that both gene panel size and transcript quantification specificity substantially influence the sensitivity and robustness of biological pathway detection.

These findings have been incorporated into the revised 'Results' section under 'Evaluation of spatial clustering and spatial pathway enrichment' and the 'Methods' section under 'Enrichment of interferon gamma pathway'.

Response Figure 2.1. Gene panel size and detection specificity impact pathway-level sensitivity in spatial transcriptomics.

a. CODEX image of OV tissue showing selected T cell-enriched (white boxes) and T cell-depleted (yellow boxes) regions, each 500 × 500 µm in size.

b. Enrichment of the interferon gamma response pathway in T cell-enriched regions across four spatial transcriptomics platforms using GSEA. NES: normalized enrichment score.

c. Total number of genes included in each platform-specific panel. Panel labels: Visium HD FFPE (Visium HD Whole Transcriptome Panel), CosMx 6K (CosMx Human 6K Discovery RNA Panel), CosMx 1K (CosMx 1000-plex Human Universal Cell Characterization Panel), CosMx IO (CosMx 100-plex Human Immuno-Oncology Panel), Xenium 5K (Xenium Prime 5K Human Pan Tissue & Pathways Panel), Xenium

Multi-Tissue (Xenium Human Multi-Tissue and Cancer Panel), and Xenium IO (Xenium Human Immuno-Oncology Panel).

d. NES and adjusted P values for interferon gamma pathway enrichment in T cell-enriched regions across different gene panels, using spatial transcriptomics data from Visium HD FFPE.

e-f. Box plots showing NES and adjusted P values for pathway enrichment in T cell-enriched regions after random downsampling of housekeeping genes (e) or interferon gamma response pathway (f). X-axis indicates the number of genes retained. Each condition was repeated 100 times.

g. Bar plot showing the effect of increasing levels of Gaussian noise on pathway enrichment scores, simulating reduced expression specificity.

2. Variations in Sensitivity, Specificity, and Concordance Across Cancer Types

The manuscript reports differences in sensitivity, specificity, and concordance across different cancer types. A more detailed discussion of these findings would provide valuable insights into their implications.

Response: We sincerely thank the Reviewer for this insightful comment. Differences in gene panels and sample variation contributed to the sensitivity discrepancy of the two imaging-based spatial transcriptomics platforms across different cancer types. When restricting the analysis to shared genes, Xenium 5K consistently detected more genes and transcripts per bin across all three tissue types (**Response Figure 1.2**). Regarding specificity, although Visium HD FFPE exhibited more pronounced transcript diffusion than Stereo-seq v1.3 in HCC, it showed better overall diffusion control across the three cancer types. This variability may arise from technical fluctuations during sample processing (**Fig. 2e and Supplementary Fig. 5g-h**). As for concordance with CODEX, platform performance varied by tissue type, primarily due to the considerable background noise in the raw CODEX signals (**Response Figure 1.4a-b**). In the updated analysis, we instead used CODEX-derived cell type annotations as reference to compute correlations with spatial transcriptomics data, which eliminated the previously observed discrepancies (**Response Figure 1.4c-f**). Furthermore, to improve spatial clustering consistency, we removed channels with regional noise likely caused by experimental artifacts (**Response Figure 1.6**). These refinements led to more consistent results and offered improved guidance for spatial transcriptomics platform selection. As suggested by the Reviewer, we have revised the 'Discussion' section to provide a clearer and more concise conclusion. The reorganized and added contents now read as:

(1) Sensitivity, specificity, and concordance with CODEX for different ST platforms

"At the technical level, both sST and iST platforms demonstrated distinct strengths and limitations. Among the sST platforms, Visium HD FFPE outperformed Stereo-seq v1.3 in transcript detection sensitivity and specificity, likely due to its targeted transcript capture strategy and reduced transcript diffusion. Within the iST platforms, Xenium 5K exhibited stronger gene-wise correlations with scRNA-seq and more effective

background signal control than CosMx 6K, which showed elevated signals for low-abundance genes. Notably, both Xenium 5K and Visium HD FFPE exhibited higher spatial concordance with CODEX-based protein maps, underscoring their spatial accuracy. Beyond transcript-level assessments, we also benchmarked cell segmentation performance. Xenium 5K achieved superior segmentation accuracy, minimizing transcript spillover between adjacent cells and enabling better resolution of single-cell boundaries. Importantly, it also captured more irregular cell morphologies, reflecting the well-established morphological heterogeneity among different cell types”.

(2) Guiding ST platform selection based on sensitivity, specificity, and concordance metrics

“Our findings provide practical guidance for selecting ST platforms based on study objectives. For single-cell level analyses, iST platforms are preferable due to their use of multichannel staining and full-cell segmentation, which enable more accurate cell boundary delineation and reduce transcript spillover. Their targeted capture strategies also enhance the sensitivity and specificity of marker gene detection, supporting cell state inference. Among the iST platforms, CosMx 6K was more susceptible to background noise, which may lead to false detection of low-abundance genes and reduce annotation fidelity. In contrast, for spatial analyses focused on tissue-region level patterns, sST platforms may be more suitable due to their broader gene coverage, which improves sensitivity for pathway-level enrichment analyses. Within this category, Visium HD FFPE offered better performance, driven by its reduced transcript diffusion and high detection accuracy. Finally, for applications involving host–microbe interactions, Stereo-seq v1.3 holds unique advantages due to its unbiased poly(A)-based capture, which allows for the detection of both human and non-human transcripts”.

Minor points and recommendations

1. Figure 1b: VISIUM HD appears to be from a different tissue section compared to CosMx and Xenium. Is this correctly analyzed as a serial section? If possible, analyses should be conducted using tissue sections with more similar HE staining patterns to ensure comparability.

Response: We thank the Reviewer for the helpful comment. For FFPE samples, we used consecutive tissue sections; however, due to the smaller capture area of the Visium HD FFPE chip (6.5 × 6.5 mm) compared to CosMx 6K (20 × 15 mm) and Xenium 5K (10.45 × 22.45 mm), only a portion of the full section was profiled, leading to the differences in H&E staining patterns observed in Figure 1b. To mitigate this issue, we performed image registration across Visium HD FFPE, CosMx 6K, and Xenium 5K staining images, mapped cells to a common coordinate system, and applied two strategies to directly compare sensitivity across platforms: (1) analyzing the commonly captured areas (**Supplementary Fig. 2a**), and (2) selecting ten tumor regions with similar morphology (**Supplementary Fig. 2c**).

2. Figure 1g: When discussing sequencing-based sensitivity, sequencing saturation is an important parameter and should be presented.

Response: We thank the Reviewer for the helpful comment. Sequencing saturation is a key metric indicating the proportion of redundant reads and often used to compare capture efficiency of different platforms. We downsampled reads from ten tumor regions to assess sequencing saturation across different depths. Visium HD FFPE exhibited higher sensitivity for human transcripts due to its targeted design. However, when transcripts from other species were also included, the difference between the two platforms was substantially reduced (**Response Figure 2.2**). We have updated Figure 1g and the corresponding 'Results' section to reflect these findings, which now reads as: "To account for differences in sequencing depth, we performed read downsampling and calculated the mean sequencing saturation across the ten selected ROIs from HCC and OV samples. While Visium HD FFPE showed lower sequencing saturation for human transcripts at comparable sequencing depths, this advantage was largely attenuated when considering all transcripts (**Fig. 1g**)".

Response Figure 2.2. Sequencing saturation analysis across platforms.

Relationship between total read counts and sequencing saturation for Stereo-seq v1.3 and Visium HD FFPE across ten morphology-matched regions. Analyses are shown separately for human transcripts only (left) and for all transcripts (right), including potential non-human or unannotated sequences.

3. Figure 2i: The manuscript states: "In COAD, we identified TLS-like structures and analyzed the spatial distribution of associated transcripts. Concordance between ST and CODEX data for MS4A1 (B cell marker) and CD3E (T cell marker) suggests that ST platforms could recapitulate the cellular organization in TME (Fig. 2i)." However, TLS structures do not appear to be clearly depicted in Stereo-seq or CosMx. If the claim is that all methods detect TLS similarly, objective numerical data should be provided to support this statement.

Response: We thank the Reviewer for raising this important point. As not all platforms demonstrated clear concordance with the adjacent CODEX sections in identifying TLS-like structures, we have revised the corresponding statement in the manuscript to more

accurately reflect these findings. The updated text in the 'Results' section now reads as: "In COAD, we identified TLS-like structures and examined the spatial distribution of corresponding transcripts. Visium HD FFPE and Xenium 5K demonstrated strong spatial concordance with CODEX for B cell, CD4⁺ T cell, and CD8⁺ T cell markers (**Supplementary Fig. 7a**)".

Reviewer #3:

Summary of the findings

Ren et al present an impressive body of work entitle Systematic Benchmarking of High-Throughput Subcellular Spatial Transcriptomics Platforms. The authors have meticulously benchmarked spatial transcriptomics platforms at a time when these technologies are starting to become more widely adopted, especially in translational research. Of note is that the authors conducted the benchmarking on three different tumor samples obtained from patients, and great care was taken to conduct sample preparation at a similar time frame. This greatly enhances the validity of the findings, since benchmarking on publicly available datasets provided by manufacturers are often from different samples which introduces a large degree of variability. In order for the work to be publishable, we recommend the authors revise the manuscript according to the concerns raised below.

Response: We thank the Reviewer for providing a concise summary of our work. As outlined in our responses below, we have carefully addressed the concerns raised. The Reviewer's constructive feedback has greatly contributed to improving the clarity, rigor, and overall quality of the revised manuscript.

Major concerns and limitations

1. The title should indicate that the benchmarking was done on human cancer samples.

Response: We thank the Reviewer for this helpful suggestion. We have revised the title to "Systematic Benchmarking of High-Throughput Subcellular Spatial Transcriptomics Platforms Across Human Tumors".

2. In the introduction, more detail about probe number in current technologies can be provided, and the advance to increase to 5k and 6k in new technologies.

Response: We thank the Reviewer for this valuable suggestion. We have expanded the 'Introduction' section to include additional context on the number of probes used in earlier spatial transcriptomics platforms and highlighted how the recent expansion to 5K and 6K gene panels represents a significant technological advancement.

The revised 'Introduction' now includes the following sentence: "Compared to earlier iST platforms such as ISS (39 genes), MERFISH (1,000 genes), and STARmap (1,020 genes), the substantially expanded gene panels of CosMx 6K and Xenium 5K offer

enhanced resolution of cellular states, enable more comprehensive inference of intercellular communication networks, and allow for broader coverage of signaling pathway activities. Notably, the increased transcriptomic coverage also supports cross-disciplinary investigations, facilitating integrative analyses across domains such as immunology, oncology, and neuroscience”.

3. Access to the raw data is not provided. It is becoming a standard that data be made available to view via online repositories (such as BioImageArchive) for accurate review. Although the SPATCH viewer the authors have created is impressive, the H&E tile scans are compressed to such an extent that histopathological interpretation is made impossible. Raw data download from SPATCH is also locked until publication.

Response: We thank the Reviewer for this important suggestion. We have now deposited the imaging data in the BioImageArchive (<http://ebi.ac.uk/biostudies/bioimages/studies/S-BIAD1900>). Furthermore, to facilitate transparency, reproducibility, and broader data reuse, we have substantially expanded the data availability in both the revised manuscript and the SPATCH website. Raw sequencing data have been deposited in the Genome Sequence Archive (<https://ngdc.cncb.ac.cn/gsa-human/browse/HRA011129>). SPATCH now offers comprehensive access to additional raw data, including (1) Stereo-seq v1.3 and Visium HD FFPE data at original resolution; (2) multi-channel immunofluorescence images, transcript-level coordinate files, and original cell segmentations for CosMx 6K and Xenium 5K. We have enabled full data download through the SPATCH viewer (<https://spatch.pku-genomics.org/#/download>). Although the paper has not yet been officially published, the dataset has already been downloaded over 2,280 times, underscoring its potential impact and broad utility to the research community.

4. Since the database the authors have created has such immense value to the field, there could be more discussion on the use of SPATCH, and how it could be leveraged to unlock current limitations in the field.

Response: We thank the Reviewer for this thoughtful suggestion. We have expanded the ‘Discussion’ section to further elaborate on the potential applications of SPATCH, emphasizing its potential utility in supporting the development of spatial transcriptomics models and in accelerating biological discovery through interactive data exploration and integration, which reads as:

“To further facilitate data exploration and reuse, we developed the SPATCH portal (<https://spatch.pku-genomics.org/>), which enables interactive visualization of gene expression and annotated cell type distributions across platforms and samples. The portal also provides access to both raw and processed ST and proteomic datasets, transcript-level spatial coordinate files, high-resolution histological images, multiplexed immunofluorescence-stained morphology images, and segmentation masks derived from both platform-specific pipelines and expert-curated nuclear annotations. Together,

these publicly available resources offer a valuable foundation for advancing spatial transcriptomics modeling, methodological innovation, and biological discovery”.

5. No information is provided in the supplementary on the patients from which the samples are pertained (age, sex, comorbidities etc.)

Response: We thank the Reviewer for this helpful suggestion. We have included the metadata in the Supplementary Table 1. To further enhance accessibility, the metadata is also available for download via the SPATCH viewer (<https://spatch.pku-genomics.org/#/download>).

6. Figure 1: Visium FFPE and Xenium 5K are serial sections but the tissue looks very different. However, Xenium 5K and CosMx 6K are serial sections and do look serial. Is the scale different? What is causing this discrepancy? The CODEX data with anti-EpCAM antibody should also be included here.

Response: We thank the Reviewer for this helpful observation. For FFPE samples, we used consecutive tissue sections; however, due to the smaller capture area of the Visium HD FFPE chip (6.5×6.5 mm) compared to CosMx 6K (20×15 mm) and Xenium 5K (10.45×22.45 mm), only a portion of the full section was profiled, leading to the differences in H&E staining patterns observed in Figure 1b. To mitigate this issue, we performed image registration across Visium HD FFPE, CosMx 6K, and Xenium 5K staining images, mapped cells to a common coordinate system, and applied two strategies to directly compare sensitivity across platforms: (1) analyzing the commonly captured areas (**Supplementary Fig. 2a**), and (2) selecting ten tumor regions with similar morphology (**Supplementary Fig. 2c**).

Additionally, as suggested by the Reviewer, we have now incorporated the CODEX data with anti-Pan-Cytokeratin antibody staining to illustrate the spatial localization of epithelial cells, facilitating more direct and consistent comparisons across platforms (**Response Figure 3.1**).

Response Figure 3.1. Spatial localization of epithelial cells in COAD sections.

H&E staining and EPCAM expression from spatial transcriptomics data, along with PanCK staining from adjacent CODEX sections of COAD samples across the four spatial transcriptomics platforms. Color intensity reflects the transcript count per 8 μm bin. Scale bars, 1 mm.

7. Line 164: The authors don't elaborate on why specifically 8- μm resolution was used for comparison when conducting comparisons for lineage markers.

Response: We thank the Reviewer for the comment. We selected the 8 \times 8 μm bin size for transcript-level analyses based on both technical and biological considerations. First, to ensure consistency across spatial transcriptomics platforms with varying resolutions, we aggregated the original transcript coordinates into 8 μm bins. Furthermore, this aggregation mitigated the issue of data sparsity inherent at higher resolutions (e.g., 1–2 μm), thereby improving the statistical robustness of downstream quantification. This bin size is also recommended by the Visium HD FFPE manufacturer as a starting resolution for the downstream analysis. Second, from a biological perspective, an 8-10 μm bin approximates the size of a single human cell, and is commonly used in spatial transcriptomics studies as a surrogate for single-cell resolution—particularly for lymphocytes which are smaller than other cells. This resolution reflects a biologically meaningful unit that balances spatial specificity with transcript detection sensitivity, facilitating cross-platform comparisons of cell type-specific marker expression. We have revised the manuscript to explain on this selection, which now reads as: “To ensure consistent resolution across platforms and balance spatial specificity with transcript detection sensitivity, all subsequent bin-level analyses were conducted at 8 μm resolution—a biologically meaningful unit approximating the typical diameter of small immune cells”.

8. Line 223: Further explanation should be given regarding the sensitivity benchmarking of sST technologies. Is the detection of non-human transcripts related to sequencing errors?

Response: We thank the Reviewer for the comment. The detection of non-human transcripts is not due to sequencing errors, but rather to the unbiased poly(A)-based capture strategy employed by Stereo-seq v1.3. This approach enables the detection of microbial or other non-human transcripts with poly(A) tails that may be present in the sample. To clarify this point, we have updated the manuscript, which now reads: “While Visium HD FFPE showed lower sequencing saturation for human transcripts at comparable sequencing depths, this advantage was largely attenuated when considering all transcripts (**Fig. 1g**). These findings suggest that the observed discrepancy is primarily attributable to Visium HD FFPE's targeted capture of the human transcriptome, in contrast to the untargeted, poly(A)-based approach of Stereo-seq v1.3, which captures both human and non-human transcripts”.

9. Assessing the number of negative probes in necrotic regions can be done to more

accurately validate non-specific binding. More negative probes would be expected here.

Response: We thank the Reviewer for the insightful suggestion. To address this concern, we re-analyzed necrotic regions in the OV sample by selecting ten $250 \times 250 \mu\text{m}$ areas and quantifying the average signal per negative control. The results showed that Xenium 5K exhibited significantly lower background signal compared to CosMx 6K (**Response Figure 3.2**), indicating reduced non-specific binding and improved signal specificity.

These changes are now reflected in the 'Results' section, which reads: "Moreover, negative control signals in the necrotic regions of OV samples were markedly reduced in Xenium 5K, reflecting lower background noise (**Supplementary Fig. 5c, d**)".

Response Figure 3.2. Detection of negative control signals in necrotic regions across CosMx 6K and Xenium 5K platforms.

a. H&E-stained images of OV tissue sections profiled by CosMx 6K and Xenium 5K. Red boxes indicate ten manually selected necrotic regions ($250 \times 250 \mu\text{m}$ each) used for background signal analysis.

b. Bar plot showing the average number of signals detected per negative control within the selected necrotic regions. "NegCode" refers to sequences not included in the decoding codebook; "NegProbe" refers to probes targeting non-human transcripts.

10. For the evaluation of marker gene detection (Supp Figure 2d), the COAD data should be included in this comparison regardless of the irregular organization of cancer cells.

Response: We thank the Reviewer for the suggestion. We selected 10 regions within the COAD sections for each dataset and calculated the average expression level of *EPCAM* per bin (**Response Figure 3.3a**). Our results revealed that Xenium 5K exhibited the highest sensitivity in detecting *EPCAM* (**Response Figure 3.3b**). As suggested by the Reviewer, we have incorporated this analysis into the manuscript, which now reads as: "To further reduce variability in the scanning areas, we selected ten regions of interest (ROIs, $400 \times 400 \mu\text{m}$ each), primarily composed of cancer cells with similar morphology and cell density from each dataset (**Supplementary Fig. 2c**). Within these ROIs, we evaluated the sensitivity of cancer cell marker genes and found that Visium HD FFPE outperformed Stereo-seq v1.3, while Xenium 5K showed higher

sensitivity than CosMx 6K (**Supplementary Fig. 2d**).”

Response Figure 3.3. EPCAM transcript detection in morphologically matched tumor regions across platforms.

a. H&E staining of Stereo-seq v1.3, Visium HD FFPE, CosMx 6K, and Xenium 5K COAD sections. Ten tumor regions with similar morphology ($400 \times 400 \mu\text{m}$ each) were selected (red boxes). Magnified views of one representative region per platform are shown below. **b.** Bar plot showing the average EPCAM transcript count per bin across the selected regions. Error bars represent the SEM.

11. The variability in the Visium HD FF data is concerning, especially if there is a possibility of RNA degradation in the samples. There, it is strongly recommended that the Visium HD FF data is not included in the study.

Response: We thank the reviewer for the insightful comment. Visium HD FF was launched after Stereo-seq v1.3 and was not included in the initial experimental design, therefore, multiple rounds of tissue sectioning were performed for the FF tissue blocks: one for Stereo-seq v1.3, one for the Visium HD FF pilot experiment, and another for the final Visium HD FF experiment. The repeated freeze–thaw cycles may lead to partial sample degradation. In contrast, the FFPE samples were sectioned simultaneously for all three platforms, thereby avoiding this issue and ensuring consistency across experiments. As suggested by the Reviewer, we have removed the Visium HD FF data from the analysis in the revised manuscript.

12. Line 218-219: The conclusion that a high proportion of reads with barcodes not matching the mask file indicates a more significant loss of spatial information (Supplementary Fig. 4e) is not entirely accurate, since the mask file in this case was only the nuclear stain and the unmatched barcodes could be from the cytosolic component of cells.

Response: We apologize for the confusion regarding the use of "mask" throughout the manuscript. In this context, "mask" does not refer to cell or nuclear segmentation

masks, but rather to the Stereo-seq chip mask file, which contains all known CID (coordinate ID) sequences. Each CID corresponds to a unique spatial position on the chip and serves an analogous role to the spatial barcodes used in Visium HD FFPE, enabling the mapping of sequencing reads to their spatial locations.

We have clarified this point in the revised manuscript, which now reads: “Stereo-seq v1.3 exhibited a higher proportion of reads passing Unique Molecular Identifier (UMI) quality control compared to Visium HD FFPE and scRNA-seq, indicating improved retention of valid transcript information (**Supplementary Fig. 4e**). However, it also showed an elevated proportion of reads with invalid spatial barcodes, indicating greater loss of spatial information (**Supplementary Fig. 4e**)”.

13. Line 244-246: Extraction of high-quality calls and re-evaluating the negative control ratio was only conducted for CosMx 6K, the mitigation of bias for low-quality signals should also be conducted for Xenium 5K (Supplementary Fig. 5c).

Response: We thank the Reviewer for this important suggestion. We re-evaluated the negative control signals in the Xenium 5K data under a range of quality control thresholds. Across all conditions, Xenium 5K consistently demonstrated substantially lower background signal levels compared to CosMx 6K (**Response Figure 3.4**), highlighting its robustness to noise irrespective of data stringency.

These changes are reflected in the ‘Results’ section, which reads: “Across a range of quality control thresholds, Xenium 5K consistently maintained a lower proportion of negative control signals, highlighting its advantage in minimizing background noise (**Supplementary Fig. 5e**)”.

Response Figure 3.4. Negative control signal levels across quality control thresholds in CosMx6K and Xenium 5K datasets.

Proportions of negative control probes (left) and negative control codes (right) across a range of quality control thresholds of the CosMx 6K and Xenium 5K platforms. X-axis indicates the fraction of retained transcript calls after applying increasing quality control stringency. Y-axis indicates the log₁₀-transformed proportion of negative control signals relative to total transcript calls.

14. The concordance variation observed between ST and CODEX could partially be described by the slight difference in cellular distribution of the adjacent section. There could also be artifacts introduced due to 'warping' of pixels during alignment. It would enhance the confidence of the results if the registered overlays of these sections are shown in the supplementary.

Response: We thank the Reviewer for the insightful comment. As the Reviewer suggested, we overlaid the registered DAPI channel from CODEX with the corresponding spatial transcriptomics images. Overall, the two modalities showed good alignment, although slight shifts were observed between adjacent tissue sections (**Response Figure 3.5**). The inter-section variability in cell type distribution might also contribute to the observed discrepancies. To account for such technical and biological variations, we employed a multi-scale grid-based analytical framework ranging from 100 μm to 500 μm . Larger grid sizes inherently mitigate the influence of local misregistration and subtle differences in cellular architecture between adjacent sections, thereby enhancing the robustness of spatial concordance assessments. Thus additional factors likely contributed to the observed variation in spatial concordance. Upon careful examination, we found that this variation was primarily driven by differences in background noise in the CODEX data across tissue types (**Response Figure 1.4a-b**). When raw CODEX signal intensities were replaced with cell counts derived from CODEX-based cell segmentation in the correlation analysis, platform variation across tissue types was substantially reduced (**Response Figure 1.4c-f**).

Response Figure 3.5. Cross-modality image registration between CODEX and spatial transcriptomics data across platforms.

For each group, the top-left panel shows the DAPI channel from CODEX, the bottom-left panel shows the corresponding morphology image from spatial transcriptomics data, and the right panel shows the merged overlay of CODEX and spatial transcriptomics images after registration. For Stereo-seq v1.3 and Visium HD FFPE, the morphology image corresponds to a grayscale-converted H&E image, whereas for

CosMx 6K and Xenium 5K, it derives from the DAPI fluorescence channel.

15. Line 285: The conclusion that iST platforms exhibited better concordance with CODEX than sST platforms (Supplementary Fig. 6c) does not have statistical merit. Visually there is more concordance in terms of structural representation, but this is not reflected in the correlation analysis.

Response: We thank the Reviewer for pointing this out. In our previous analysis, we observed substantial variability in the concordance between spatial transcriptomics platforms and CODEX data across different tissue types. Further investigation revealed that this variation was primarily driven by the background noise of the raw CODEX signal. To address this, we revised our correlation analysis by replacing the raw signal intensities with the abundance of annotated cell types inferred from CODEX data. Additionally, we refined the definition of cell-type-specific signatures to enhance the robustness of the cross-platform comparisons. The top 15 differentially expressed genes for each cell type were identified from matched scRNA-seq data in each tissue type, and only those present in at least two tissue types were retained to ensure cross-tissue consistency. To minimize the impact of platform-specific gene panel differences, the resulting gene set was further intersected with the common genes shared by all four spatial transcriptomics platforms. Compared to the previous approach that relied on the top 50 differentially expressed genes, the updated method utilized genes with stronger differential expression, thereby reducing contamination from transcripts originating from other cell types. This refinement yields cell-type signatures with improved specificity and enhances the accuracy of cross-platform comparisons (**Response Figure 3.6a**). Using the updated method, we calculated the correlation coefficients between spatial transcriptomics and CODEX data within the selected ROIs (**Response Figure 3.6b**). Imaging-based spatial transcriptomics platforms showed comparable, rather than superior, concordance with CODEX relative to sequencing-based platforms. Accordingly, we have removed this statement from the revised manuscript.

Response Figure 3.6. Spatial correlation between transcriptomic signatures and CODEX-inferred immune cell distributions across platforms.

a. Representative immune-enriched regions from COAD samples profiled using all four spatial transcriptomics platforms. Left to right: H&E-stained histology, spatial distribution of transcriptomic signature scores for B cells, CD4⁺ T cell, and CD8⁺ T cells derived from spatial transcriptomics data, and corresponding CODEX staining images (DAPI, CD20, CD8, CD4). For spatial transcriptomics data, color intensity represents the corresponding signature score of each 8 × 8 μm bin.

b. Spearman correlation between spatial transcriptomic signature scores and CODEX-quantified cell densities per spatial bin for B cells, CD4⁺ T cells, and CD8⁺ T cells within the regions shown in panel **a**.

16. Paragraph 288 – 295 seems out of context. Further detail should be provided as to the relevance of this finding?

Response: We thank the Reviewer for the helpful comment. To better illustrate the concordance between spatial transcriptomics platforms and CODEX in regions with specialized tissue architecture across different cancer types, we have reorganized the relevant section and integrated this paragraph into the earlier analysis of gene expression in TLS-like regions. The revised paragraph now reads:

“We first assessed local concordance between ST and CODEX data in representative tissue areas. Tertiary lymphoid structures (TLS), aggregates of T and B cells that support both humoral and cellular immunity, play a crucial role in anti-tumor responses⁶. In COAD, we identified TLS-like structures and examined the spatial distribution of corresponding transcripts. Visium HD FFPE and Xenium 5K demonstrated strong spatial concordance with CODEX for B cell, CD4⁺ T cell, and CD8⁺ T cell markers (**Supplementary Fig. 7a**). In HCC, we leveraged the liver’s intricate vascular

architecture to evaluate the localization of *CD34* (endothelial cell marker) near vascular structures. Xenium 5K showed high levels of *CD34* transcripts along vascular edges (**Supplementary Fig. 7b**). In OV, we identified macrophage-rich regions and assessed *CD68* transcript localization, with all platforms detecting substantial *CD68* expression (**Supplementary Fig. 7c**)”.

17. The cell segmentation results are hard to interpret. Image quality for Fig 3a is very low, and given that cell boundary stains were used, this should be included in the image panel, along with the boundary segmentation to more clearly represent the differences.

Response: We thank the Reviewer for this insightful comment. To improve clarity and interpretability of the segmentation results, we have revised Figure 3a by enlarging the field of view to $250 \times 250 \mu\text{m}$, allowing clearer visualization of the automatically segmented cell boundaries and manually labeled nuclear boundaries (**Response Figure 3.7a**). Moreover, as suggested, we have incorporated the membrane staining images for both CosMx 6K and Xenium 5K corresponding to the same regions shown in the segmentation overlays (**Response Figure 3.7b**). These additions enable a more direct comparison between the membrane signal and the automatically segmented cell boundaries. We believe this enhancement provides better visual context for assessing segmentation accuracy, particularly in relation to membrane-defined cell borders.

Response Figure 3.7. Evaluation of cell segmentation accuracy across platforms.

a. Comparison of automatic and manual segmentation in $250 \times 250 \mu\text{m}$ regions from HCC sections profiled by Stereo-seq v1.3, CosMx 6K, and Xenium 5K. Left: automatically annotated cell boundaries. Middle: manually annotated nuclear masks. Right: overlay of the two annotations, with white polygons denoting automatic annotations and blue/red filled masks indicating manual annotations.

b. Membrane staining and whole-cell segmentation boundaries in matched $250 \times 250 \mu\text{m}$ regions from CosMx 6K and Xenium 5K HCC sections. Left: raw fluorescence images with membrane signal. Right: corresponding segmentation

overlays displaying full-cell boundaries.

18. Cell segmentation stains for Xenium are not specified in the methods.

Response: We thank the Reviewer for pointing this out. We have revised the 'Methods' section to specify the segmentation kit used for Xenium.

The revised section now reads: "The Xenium Cell Segmentation Staining Reagents (PN-1000661, 10x Genomics) were used for membrane, cytoplasm and nuclear staining. The assay was performed using the Xenium Prime 5K Human Pan Tissue & Pathways Panel (PN-1000724, 10x Genomics), which targets 5,001 individual human genes".

19. It appears that the manual segmentation was only conducted on nuclei and not on entire cells? Representative Fig 3a does not make this clear. No mention or description is given in the methods on how manual segmentation was done or in which software. Was this done by thresholding or manually drawing outlines? In the methods, it is specified that the ST and CODEX correlations made use of DAPI for 'cell segmentation' using StarDist and QuPath. Was this the manual approach referred to for the comparison in Fig 3 as well? Can this truly be called 'ground truth'?

Response: We thank the reviewer for the detailed comments.

(1) The manual segmentation presented in Figure 3a was performed on nuclei, not entire cells. Stereo-seq v1.3 platform used H&E staining, which did not resolve cell membranes and only allowed for identification of nuclei. To ensure a fair and consistent comparison across platforms, we manually annotated nuclear boundaries in the selected regions for all three platforms. Moreover, the membrane staining quality varied considerably between Xenium 5K and CosMx 6K, and nuclear boundaries were generally more clear and consistent across platforms. Thus, manual nuclear segmentation provides a more robust and unbiased reference for evaluating segmentation performance.

(2) Manual nuclear segmentations were generated using the Labelme software. Nuclear contours were manually delineated by experienced pathologists based on DAPI or H&E signals. This approach ensures high-quality reference annotations grounded in expert interpretation of histological and fluorescence signals.

(3) For CODEX, we first applied StarDist¹ to segment nuclei from the DAPI channel, and then expanded the nuclear masks by 5 μ m to approximate whole-cell boundaries for downstream analyses. The manual labeling used in the CODEX annotation context refers solely to cell type annotations, not to manual delineation of cellular or nuclear boundaries. Therefore, the segmentation of CODEX data used for correlation analysis is unrelated to the manual segmentations in Figure 3a.

Taken together, we believe the manually annotated nuclei used in Figure 3a provide a reasonable and robust ground truth for assessing segmentation performance across platforms. To avoid confusion, we have updated the figure legend and 'Methods'

section to explicitly clarify the source and scope of manual and automated segmentation data used in each analysis.

Figure legend:

“Comparison of platform-derived automatic cell segmentation and manual nuclear segmentation across Stereo-seq v1.3, CosMx 6K, and Xenium 5K HCC sections. For each platform, a 250 × 250 μm region is shown. Left, automatically annotated cell boundaries; middle, manually annotated nuclear boundaries; right, overlay of automatic and manual annotations, with white polygons denoting automatic annotations and blue/red filled masks indicating manual annotations. Scale bars, 50 μm”.

Methods section:

“Manual segmentation was conducted using the software Labelme, where nuclear outlines were drawn manually based on DAPI and H&E staining”.

20. Line 313: iST platforms had a higher proportion of transcripts confined within the segmented cells compared to the sST platform (Supplementary Fig. 7f). Authors should clearly state whether this was comparing whole cell segmentation or nuclei segmentation.

Response: We thank the Reviewer for the comment. For this comparison, Stereo-seq v1.3 utilized nuclear segmentation with a 5 μm radial expansion beyond the H&E-defined nuclei, whereas Xenium 5K and CosMx 6K utilized whole-cell segmentation as provided by their respective platform pipelines. To improve clarity and prevent confusion, we have explicitly distinguished between nucleus-based and whole-cell segmentation approaches in the revised manuscript.

Results section:

“CosMx 6K and Xenium 5K performed cell segmentation based on multi-channel staining images that included nuclear, membrane, and cytoplasmic markers, while Stereo-seq v1.3 estimated cell boundaries by expanding nuclear masks by 5 μm. Following cell segmentation, we compared transcript and gene counts within segmented cells across platforms. CosMx 6K and Xenium 5K retained a higher proportion of transcripts within the cell boundaries (**Supplementary Fig. 10a**), indicating more effective transcript assignment.”

21. Throughout the entire paper, the term cell segmentation is used, even though it is often nuclei segmentation. This should be made consistent and clear so as to prevent confusion.

Response: We thank the Reviewer for pointing this out. In the revised manuscript, we have clarified the distinction between nuclear segmentation and whole-cell segmentation and have ensured consistent use of the appropriate terminology

throughout the manuscript to avoid confusion.

22. Although official segmentation tools are not provided by the supplier for Visium HD FFPE, the existence of many open source segmentation tools still make it possible to do a fair comparison. Since H&E images are available, simple cell segmentation in QuPath could be applied to give an indication of how the transcript counts compare to other technologies.

Response: We thank the Reviewer for the insightful suggestion. As suggested by the Reviewer, to ensure consistent segmentation across platforms, we performed nucleus segmentation using the open-source tool StarDist for all four spatial transcriptomics platforms across three cancer types. Our analysis revealed that Xenium 5K consistently detected a higher number of genes and transcripts per segmented nucleus, indicating its superior sensitivity under standardized segmentation conditions (**Response Figure 3.8**).

These updates are now reflected in the revised 'Results' section, which reads: "Additionally, we performed whole-slide nuclear segmentation using StarDist¹ across the four ST platforms. Xenium 5K consistently identified nuclei with higher gene and transcript counts, both for all detected genes and for genes shared across platforms (**Supplementary Fig. 10c**)".

Response Figure 3.8. Comparison of transcript and gene counts per cell/nucleus across platforms

Log₂-transformed transcript and gene counts per nucleus are shown for COAD, HCC, and OV samples. For scRNA-seq data, each data point represents a single cell. For spatial transcriptomics platforms, each data point corresponds to a nucleus segmented by StarDist. Analyses were performed using both all detected genes (left) and common genes shared across platforms (right).

23. Line 346: Evaluation of Cell Clustering. In addition to showing the clustering visually (Figure 4d), a quantitative metric should be used to compare the accuracy. Perhaps a Jacard index or similar. The visual only comparison is hard to interpret.

Response: We thank the Reviewer for the comment. We would like to clarify that Figure

4d presents spatial cell type annotations. These annotations were generated by transferring cell type labels from matched scRNA-seq datasets to the spatial transcriptomics data using multiple automated annotation tools, rather than relying on conventional unsupervised clustering followed by manual annotation. To enable quantitative cross-platform comparison, we also performed cell type annotation on the corresponding CODEX data and evaluated spatial concordance between the two modalities (**Fig. 4e, f**).

24. Line 778: Diffusion distances – Diffusion must also be occurring within the tissue. In the discussion, it would be good to see a comparison of FFPE and Frozen in general with regards to diffusion distance.

Response: We thank the Reviewer for this thoughtful comment. For sequencing-based spatial transcriptomics platforms, both intra- and extra-tissue transcript diffusion are primarily driven by membrane permeabilization. While transcript diffusion outside the tissue can be directly observed, diffusion within the tissue is more difficult to quantify due to the lack of a gold standard. Nonetheless, spatial correlation with CODEX provides an indirect measure of intra-tissue diffusion. Specifically, lower correlations over larger spatial grids may indicate more substantial diffusion of transcripts from their cells of origin. Visium HD FFPE exhibited stronger spatial concordance with CODEX compared to Stereo-seq v1.3, suggesting reduced transcript diffusion within the tissue, which is consistent with our observations of lower extra-tissue diffusion.

Among the factors contributing to transcript diffusion, permeabilization is the most critical, regardless of whether the sample is FF or FFPE. Both the composition of the permeabilization reagent and the duration of treatment influence the extent of diffusion, with over-permeabilization increasing the risk of transcript leakage⁴. Additionally, in FF samples, improper or delayed freezing can result in ice crystal formation that disrupts cell membranes, further exacerbating transcript diffusion. In contrast, FFPE embedding may compromise RNA integrity, and the resulting RNA fragmentation increases the likelihood of transcript leakage during the permeabilization process. We have incorporated these considerations into the revised 'Discussion' section.

25. Optical crowding is not mentioned, but must be impacting the large 5k and 6k panels. This should be included in the discussion or introduction.

Response: We thank the Reviewer for pointing this out. Optical crowding is a well-recognized challenge for large gene panels such as the 5K and 6K panels. To mitigate this issue, both the CosMx 6K and Xenium 5K platforms adopt strategies such as reducing the number of probes per gene and increasing the number of imaging cycles. While smaller panels may exhibit slightly higher detection efficiency, a previous study has shown that the 5K panel achieves detection performance comparable to Chromium-based scRNA-seq³. We have added this clarification to the revised 'Discussion' section.

26. Line 506: We strongly suggest removing the statement about 'critical contribution to the field' - this should be implied in quality and depth of the study.

Response: We thank the Reviewer for the suggestion. We have revised the statement to focus on the specific strengths and contributions of the study, without only claiming a "critical contribution to the field."

Minor points and recommendations

1. Remove website from abstract, this should be in the main text.

Response: We thank the Reviewer for pointing this out. We have removed the website from the abstract.

2. Graphs are small to illegible.

Response: We thank the Reviewer for the suggestion. We have revised the figure layout and enlarged previously small panels for better readability.

3. No scale bars are provided on any of the images.

Response: We thank the Reviewer for pointing this out. Scale bars have been added to the relevant images.

4. Not all abbreviations are defined (for example SPATCH and UMI)

Response: We thank the Reviewer for pointing this out. We have updated the manuscript to include definitions for all abbreviations upon first use.

5. Figure 1: use same colors across all graphs (ie Fig 1 e and f)

Response: We thank the Reviewer for this suggestion. We have standardized the color scheme across all graphs, assigning a consistent color to each platform.

6. Supplementary Figure 1: At least one of the chosen ROIs should be enhanced to so it can be validated whether they are indeed representative of cancer cells with similar morphology.

Response: We thank the Reviewer for this suggestion. We have added a zoomed-in example for one of the selected ROIs to better illustrate its morphology.

7. Figure 2c is not displaying a ratio (as mentioned in text) but rather displays counts.

Response: We thank the Reviewer for pointing this out. The x-axis represents the proportion of signal from control probes, and the y-axis indicates the corresponding number of probes. We have revised the figure legend and in-figure labels to make this clearer.

8. Fig 2e: Figure legend on y-axis does not show distance metric (was it in microns or pixels?)

Response: We thank the Reviewer for pointing this out. We have corrected the y-axis

and labeled the distance in microns.

9. Figure 2g: the blue stain from the CODEX is barely visible. A using a more pronounce color would be suggested.

Response: We thank the Reviewer for pointing this out. We have adjusted the color display settings to enhance visibility.

10. Figure 4a, a legend for the clusters is missing.

Response: We thank the Reviewer for pointing this out. Colors indicate cluster identities without direct cell type annotation. We have clarified in the legend that the colors are arbitrary labels.

Reviewer #4:

References:

1. Weigert, M., Schmidt, U. & Ieee in IEEE International Symposium on Biomedical Imaging Challenges (IEEE ISBIC) (Kolkata, INDIA; 2022).
2. Stringer, C., Wang, T., Michaelos, M. & Pachitariu, M. Cellpose: a generalist algorithm for cellular segmentation. *Nat Methods* **18**, 100–106 (2021).
3. Bilous, M. et al. From Transcripts to Cells: Dissecting Sensitivity, Signal Contamination, and Specificity in Xenium Spatial Transcriptomics. *bioRxiv*, 2025.2004.2023.649965 (2025).
4. You, Y. et al. Systematic comparison of sequencing-based spatial transcriptomic methods. *Nat Methods* **21**, 1743–1754 (2024).
5. Wu, T. et al. clusterProfiler 4.0: A universal enrichment tool for interpreting omics data. *Innovation (Camb)* **2**, 100141 (2021).
6. Sautes-Fridman, C., Petitprez, F., Calderaro, J. & Fridman, W.H. Tertiary lymphoid structures in the era of cancer immunotherapy. *Nat Rev Cancer* **19**, 307–325 (2019).

Response letter to Review comments of the manuscript “Systematic Benchmarking of High-Throughput Subcellular Spatial Transcriptomics Platforms Across Human Tumors” (NCOMMS-25-06494A).

RESPONSES TO REVIEWERS

Reviewer #1:

I would like to thank the authors for their thoughtful and thorough responses to my comments. I appreciate the detailed justifications provided, the inclusion of additional analyses, and the clarifications throughout the revised manuscript. The authors have successfully addressed all my major concerns, and the quality and clarity of the work have improved substantially. I have no further comments regarding the manuscript.

We would like to sincerely express our gratitude to the Reviewer for the thoughtful and constructive feedback on our manuscript. We greatly appreciate the time and effort dedicated to reviewing our work. The insightful comments and suggestions have been invaluable in improving the quality and clarity of our study.

Reviewer #2:

Thank you for your detailed reply. Regarding Response Figure 2, I feel it does not address the core of my concern. In spatial transcriptomics, the biological pathways that play important roles in the tissue of interest are often a primary focus. Therefore, differences in gene coverage and the lower specificity of certain panels can have a substantial impact on biological pathway analysis. In my view, analyzing different locations on separate tissue slices or restricting the analysis to only overlapping genes is unnecessary. Instead, what matters is how much the list of enriched biological pathways differs when GSEA is performed on the same position across serial sections. Because spatial transcriptome data are sparse, I expect that CosMx’s lower specificity will have a large effect on pathway enrichment outcomes.

To perform this analysis, high-quality serial sections are needed. From the figures you presented, it appears that the Xenium and CosMx samples are from nearly identical positions, suggesting it should be possible to conduct GSEA at the same location for these two platforms. For other platforms, please include comparable analyses if same-position data are available. I am not looking for similar results across platforms; rather, understanding how different the enriched pathway lists are—and which pathways each method can detect more easily—would provide valuable insights to users. As such, I do not believe Response Figure 2 is essential. I would appreciate it if you could consider creating the figure I have requested, even as a supplemental figure. Regarding the other parts of your response, you have provided sufficient answers to my questions.

We sincerely thank the Reviewer for the thoughtful follow-up and constructive suggestions. As emphasized by the Reviewer, differences in gene coverage and

specificity across platforms can substantially influence biological pathway enrichment analyses, especially when evaluating matched tissue regions across serial sections. To address this point, we performed a revised analysis using regions that are anatomically aligned across serial sections processed with Visium HD FFPE, CosMx 6K, and Xenium 5K. As noted by the Reviewer, the availability of nearly identical tissue positions across these three platforms makes them well-suited for direct comparison. For each matched region, including immune-infiltrated, tumor, and normal epithelial areas (**Response Figure 1a**), we conducted differential expression analysis followed by pathway enrichment analysis, without restricting the analysis to overlapping genes, to evaluate the impact of gene coverage and specificity on downstream biological interpretation.

Our results revealed notable differences in the enriched pathway profiles across the three platforms. Xenium 5K consistently identified the highest number and proportion of differentially expressed genes (DEGs), while CosMx 6K identified the fewest number of DEGs (**Response Figure 1b-e**). These findings align with the Reviewer's expectation that lower specificity and more limited gene coverage can reduce the sensitivity and breadth of pathway detection. Gene Ontology (GO) analysis revealed that Xenium 5K consistently identified a greater number of biologically relevant pathways in both immune and tumor regions (**Response Figure 1f-g**). This suggests that its higher specificity enhances sensitivity in detecting meaningful biological processes. While a subset of enriched pathways was shared across platforms, a substantial proportion was platform-specific (**Response Figure 1h-i**), underscoring the importance of platform selection.

More specifically, analysis of the top 10 enriched pathways for each platform revealed distinct patterns. In immune-infiltrated regions, Xenium 5K prioritized immune-related pathways such as T cell activation and leukocyte-mediated cytotoxicity (**Response Figure 1j**), whereas CosMx 6K and Visium HD FFPE more frequently identified pathways related to extracellular matrix organization (**Response Figure 1j**). In tumor regions, both Xenium 5K and Visium HD FFPE were enriched for cell cycle and mitotic progression pathways, while CosMx 6K showed stronger signals for metabolic and stress-response pathways (**Response Figure 1k**). These results indicate that different platforms have varying sensitivities in detecting distinct biological pathways.

These changes are now reflected in the 'Results' and 'Methods' sections:

Results:

"To evaluate each platform's ability to resolve biologically meaningful pathways, we examined the enrichment of functional pathways across matched tissue regions. Three region types, including immune cell-infiltrated areas, tumor regions, and normal epithelial regions, were selected from anatomically aligned positions on consecutive sections processed with Visium HD FFPE, CosMx 6K, and Xenium 5K (**Supplementary Fig. 16a**). Given that reduced sensitivity and specificity may limit the detection of region-specific gene expression, we first examined differentially

expressed genes (DEGs) between immune-infiltrated and tumor regions, and between tumor and normal regions. Xenium 5K identified the highest number and proportion of DEGs across the three platforms, whereas CosMx 6K identified the fewest (**Supplementary Fig. 16b-e**). While some DEGs overlapped across platforms, many were platform-specific (**Supplementary Fig. 16d, e**), highlighting differences in gene detection sensitivity and coverage.

We next performed Gene Ontology (GO) pathway enrichment analysis using these DEGs. In immune-infiltrated regions, Xenium 5K uniquely prioritized immune-related pathways, including T cell activation and leukocyte-mediated cytotoxicity, whereas Visium HD FFPE and CosMx 6K more frequently enriched for pathways related to extracellular matrix organization (**Supplementary Fig. 16f**). In tumor regions, Visium HD FFPE and Xenium 5K identified pathways associated with cell division and proliferation, while CosMx 6K enriched for pathways associated with metabolisms (**Supplementary Fig. 16g**). Across all comparisons, Xenium 5K detected the largest number of significantly enriched pathways (**Supplementary Fig. 16h, i**). Although some pathways were shared across platforms, each platform also revealed unique enrichments (**Supplementary Fig. 16j, k**), underscoring their differing sensitivities and gene coverage profiles.”

Methods:

“To assess pathway-level differences across spatial regions, differentially expressed genes (DEGs) were identified using the scanpy Python package. Comparisons were made between immune-infiltrated versus tumor regions and tumor versus normal epithelial regions. Genes with adjusted p-value ≤ 0.05 and fold change ≥ 2 were defined as DEGs and subsequently subjected to GO enrichment analysis (R package clusterProfiler⁵⁸, v.4.6.2). Pathways with adjusted p-value ≤ 0.05 were retained for downstream comparisons across platforms.”

Response Figure 1. Comparison of regional differential gene expression and pathway enrichment across platforms.

a. Representative regions of interest (ROIs) from the COAD samples across Visium HD FFPE, CosMx 6K, and Xenium 5K platforms. ROI1 corresponds to an immune cell-rich region, ROI2 to a tumor region, and ROI3 to a normal epithelial region. The three columns on the right show magnified views of each ROI (1 mm x 1 mm). Scale bars, 1 mm.

b-c. Bar chart showing the proportion of DEGs (adjusted p-value ≤ 0.05 and fold change ≥ 2) relative to the total number of genes in each platform's panel, for immune cell-rich region vs. tumor region (**b**) and tumor region vs. normal region (**c**).

d-e. Venn diagram illustrating the overlap of DEGs identified across the three platforms, for immune cell-rich region vs. tumor region (**d**) and tumor region vs. normal region (**e**).
f-g. Bar plots displaying the total number of significantly enriched GO pathways (adjusted p-value ≤ 0.05) detected in immune cell-rich regions (**f**) and tumor regions (**g**) by each platform.

h-i. Venn diagrams showing the overlap of significantly enriched pathways across platforms in immune cell-rich regions (**h**) and tumor regions (**i**).

j-k. Lollipop plots showing the top 10 enriched GO pathways identified in immune cell-rich regions (**j**) and tumor regions (**k**). Dot colors represent the $-\log_{10}$ transformed adjusted p-values, and the x-axis indicates the ratio of DEGs involved in each pathway.

Reviewer #3

We thank the authors for applying our recommendations and amending their manuscript accordingly. In general the majority of our concerns have been addressed and additional analyses have been completed and supplementary data provided. However, we need to highlight the following inconsistencies pertaining mostly to the image data and segmentation data. We therefore recommend these minor revisions prior to publication.

We sincerely thank the Reviewer for the time and effort in reviewing our manuscript. We greatly appreciate the constructive comments, which have been instrumental in improving the quality and clarity of our work. In response to the highlighted inconsistencies, particularly regarding the image and segmentation data, we have carefully reviewed and revised the relevant figures, terminology, and associated analyses. These updates have been incorporated into the revised manuscript and supplementary materials.

1. The newly generated images in Supplementary Figure 2a for Xenium and CosMx are extremely dim and tissue outlines can't be viewed accurately. Please enhance the contrast so the tissue is actually visible.

We sincerely appreciate the Reviewer's helpful suggestion regarding visibility. We have adjusted the brightness and increased the contrast of the Xenium and CosMx images in **Supplementary Figure 2a** to ensure that tissue outlines are clearly discernible (**Response Figure 2**). We have also applied similar enhancements to **Supplementary Figure 2c** to maintain consistency and improve overall visual clarity across related figures.

Response Figure 2. Overlapping tissue regions across FFPE sections. Representative overlapping tissue regions from matched FFPE tissue sections profiled using Visium HD FFPE, CosMx 6K, and Xenium 5K for COAD, HCC, and OV samples. Tissue morphology is visualized using H&E staining for Visium HD FFPE and DAPI staining for CosMx 6K and Xenium 5K. Scale bars, 1 mm.

2. Given that Stereo-seq captures non-human transcripts so sensitively, we recommend the authors identify whether these are associated with microbial genomes to showcase that any possible contamination will be reflected in the Stereo-seq data. This will allow for a precautionary recommendation to readers/potential users that special care must be taken when conducting slide preparation for Stereo-seq specifically.

We sincerely appreciate the Reviewer's insightful suggestion. In the original version of the manuscript, we had not examined the origin of reads from Stereo-seq v1.3 that did not align to the human genome. As suggested by the Reviewer, we performed additional analysis in this revised version to assess whether these unmapped reads may originate from bacterial or viral genomes. Specifically, we used Kraken2 (v.2.1.6) to align the unmapped reads to a reference database comprising bacterial and viral genomes. Our results indicate that a small proportion of the unmapped reads aligned to bacterial (1.39%-3.02%) and viral (0.06%-1.52%) genomes (**Response Figure 3**), suggesting that contamination from non-human species is minimal in our dataset and unlikely to have a significant impact on the analysis results.

To ensure analytical consistency and biological interpretability, we have revised the manuscript to include only human transcript alignment results. The updated sentence now reads as "To account for differences in sequencing depth, we performed read

downsampling and calculated the mean sequencing saturation across the ten selected ROIs from HCC and OV samples. Visium HD FFPE showed lower sequencing saturation for human transcripts at comparable sequencing depths (**Fig. 1g**)”.

Response Figure 3. Alignment of unmapped Stereo-seq v1.3 reads to bacterial and viral genomes in HCC and OV samples. Box plot showing the percentage of transcripts – originally unmapped to the human genome – that aligned to bacterial (left) and viral (right) reference genomes in Stereo-seq v1.3 data from HCC and OV samples. Alignment was performed using Kraken2 (v2.1.6).

3. Again the representative tissue images for the CosMx and Xenium data are dim to illegible. No scale bars are present for Supplementary Image 6 either in this context. Please ensure that these are amended prior to publication.

We sincerely appreciate the Reviewer’s comment. We have revised the DAPI channel images and corresponding CODEX overlay images for both CosMx 6K and Xenium 5K to improve visibility. Specifically, we enhanced the brightness and contrast to ensure that tissue structures are clearly discernible. Additionally, we have added scale bars to the relevant panels in **Supplementary Figure 6** to provide spatial reference and improve interpretability. These updates have been incorporated into the figure.

Response Figure 4. Image alignment between ST and CODEX. For each pair of tissue sections, the top-left panel shows the DAPI channel from CODEX, the bottom-left panel shows the corresponding morphology image from the spatial transcriptomics data (CosMx 6K or Xenium 5K), and the right panel presents the merged overlay following image registration. Scale bars, 1 mm.

4. While the authors do provide updated images showing cell membranes, the merged images in Response figure 3.7a do not show an actual merged between nuclei and cell membrane segmentations. Furthermore, the term ‘annotation’ is being used here when it should be ‘segmentation’ to avoid confusion. A more zoomed in representation of outer membrane vs nuclei segmentation would be recommended for a journal of this calibre (as the authors have done in Supplementary Figure 13).

We sincerely appreciate the Reviewer’s insightful suggestions regarding the segmentation visualization and terminology. In response, we have updated the CosMx 6K and Xenium 5K multi-channel staining images to more clearly display both the automated cell membrane segmentation and corresponding manual nuclear segmentation (**Response Figure 4a**). To improve clarity and align with standard terminology, we have also replaced the term “annotation” with “segmentation” throughout the relevant sections of the manuscript.

To address the Reviewer’s comment about higher-resolution visualization, we have included zoomed-in views highlighting the alignment between nuclear and cell membrane segmentations across three platforms (**Response Figure 4b**). All changes have been reflected in the updated **Figure 3a** and **Supplementary Figure 9g**.

Response Figure 4. Comparison of cell and nuclear segmentation across ST platforms.

a. Representative segmentation results from CosMx 6K and Xenium 5K platforms using HCC tissue sections. For each platform, a $250 \times 250 \mu\text{m}$ region is shown. Top row, automatically segmented cell boundaries; middle row, manually segmented nuclear boundaries; bottom row, overlay of both. White polygons indicate automated cell segmentations, and yellow polygons indicate manual nuclear segmentations. Scale bars, $50 \mu\text{m}$.

b. Zoomed-in views of the regions shown in panel (a), highlighting segmentation detail. For each platform, the upper row shows the original images, and the lower row shows overlays of automatic and manual segmentation results. White polygons indicate automatic cell segmentations; blue and yellow polygons indicate manual nuclear segmentations.

5. We have concern regarding the inconsistency of segmentation data. In Figure 3 nuclei segmentation based efficiency is compared between technologies based on manual nuclei segmentation which makes sense due to differences between technologies. However, in the following datasets the authors then assign transcripts to whole cell segmentations based either on nuclei dilation method (expansion from StarDIST) for Stereo-seq data and then Xenium and CosMx cell boundary based segmentation. Doing comparison on segmentation efficiency on nuclei only is made void if the rest of the downstream analysis is actually done on whole cells.

Since a dilation approach is used to capture transcripts in whole cells for Stereo-seq data, the description and axis legend of Figure 10c must change from 'transcripts detected per cell/nucleus' to only 'transcripts detected per cell'. It is biologically

inaccurate to state that both cell and nuclei encapsulate the same compartment of tissue. Software version for Labelme must also be provided.

We sincerely appreciate the Reviewer's constructive feedback regarding segmentation consistency and figure labeling. In Figure 3b, we used the manually segmented nuclei as the reference standard to assess the accuracy of automated cell segmentation across platforms. To clarify, the comparison focused on the number of segmented cells, without addressing the specific boundaries of the segmentation. The automatic segmentation results used in this comparison were based on whole-cell boundaries, as shown in Figure 3a, rather than on nuclear segmentation.

We agree that the description may have caused confusion, and we have revised the corresponding text to improve clarity. The revised manuscript now reads as "The automatic cell segmentation results from CosMx 6K and Xenium 5K closely matched the number of cells manually identified by the manual nuclear segmentation within the same field of view, indicating their segmentation accuracy and reliability (**Fig. 3a, b and Supplementary Fig. 9f, g**). In contrast, Stereo-seq v1.3 exhibited reduced segmentation accuracy, likely due to staining artifacts that led to the misclassification of non-cellular structures as cells (**Fig. 3a, b and Supplementary Fig. 9f, g**)".

With respect to downstream analyses, we acknowledge the Reviewer's important point that while Figure 3 assessed segmentation efficiency based on nuclei counts, transcript assignment in later analyses was based on whole-cell segmentation. Specifically, for Stereo-seq v1.3, we utilized the platform's proprietary tool CellBin¹, which uses a nuclei dilation-based approach to approximate whole-cell segmentation.

The ST data in **Supplementary Figure 10c** was generated with nuclear segmentation method StarDist¹ to mitigate the bias induced by the platform-specific segmentation pipelines. Additionally, we included scRNA-seq data as reference to evaluate the data quality. Therefore, the unit for scRNA-seq is cell, while for ST data, the unit is StarDist-derived nucleus. Consequently, the axis label in this figure reflected both 'cell' and 'nucleus'.

Lastly, we have updated the manuscript to include the version of Labelme used for manual segmentation, which was v.5.5.0.

References:

1. Li M, *et al.* CellBin enables highly accurate single-cell segmentation for spatial transcriptomics. *bioRxiv*, 2023.2002.2028.530414 (2024).